# CodeUpdateArena: Benchmarking Knowledge Editing on API Updates

## Abstract

Large language models (LLMs) are increasingly being used to synthesize and reason about source code. The libraries and API functions they invoke are continuously evolving, with functionality being added or changing. Yet, no prior work has studied how an LLM's knowledge about code API functions can be updated. To fill this gap, we present CodeUpdateArena, a benchmark for knowledge editing in the code domain. An instance in our benchmark consists of a synthetic API function update paired with a program synthesis example that uses the updated functionality; our goal is to update an LLM to be able to solve this program synthesis example *without providing documentation of the update at inference time*. Compared to knowledge editing for facts, success here is more challenging: a code LLM must reason about the semantics of the modified function rather than just reproduce its syntax. Our dataset is constructed by first prompting GPT-4 to generate atomic and executable function updates. Then, for each update, we generate program synthesis examples whose code solutions are prone to use the update. Our benchmark covers updates of various types to 54 functions from seven diverse Python packages, with a total of 670 program synthesis examples. Our experiments show that fine-tuning open-source code LLMs (i.e., DeepSeek, CodeLlama) on documentation of a new update does not allow them to incorporate changes for problem-solving. However, prepending the same information does help, establishing that the information is present, and careful fine-tuning on examples demonstrating the update shows improvement, paving the way for better knowledge editing techniques for code.

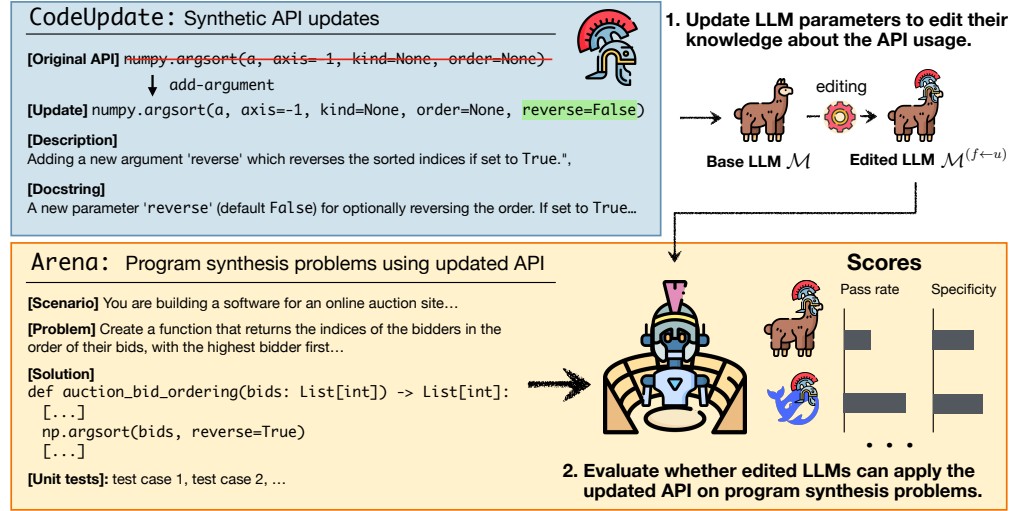

Figure 1: CodeUpdateArena overview. We generate synthetic API updates, and then evaluate whether an edited model can successfully apply the updated API on a targeted program synthesis instance.

# 1 INTRODUCTION

Large language models (LLMs) have demonstrated strong abilities to synthesize code to solve problems (Chen et al., 2021; Li et al., 2023; DeepSeek-AI et al., 2024; Guo et al., 2024a). This capability enables them to use external libraries: they can invoke standard libraries for data science-related tasks (Lai et al., 2023), program SMT solvers (Ye et al., 2023), or use external modules for tasks like computer vision (Gupta & Kembhavi, 2023). However, such APIs are not static and adherence to older APIs can cause failures. For example, in a live demo,[1] GPT-4 failed to correctly implement a Discord bot due to outdated API knowledge. To be maximally useful, LLMs for code generation need to stay in sync with API updates, even those that occur after pre-training.

A separate line of research studies knowledge editing for LLMs on simple facts. Typical use-cases here are teaching LLMs about new entities (Onoe et al., 2023), updating roles of existing entities like who the British prime minister is now (De Cao et al., 2021; Mitchell et al., 2022), and other such temporally-sensitive knowledge (Zhang & Choi, 2021). A number of techniques have been presented for these settings to efficiently update the parameters of LLMs, such as with a single gradient update (Mitchell et al., 2022; Meng et al., 2023) or with a small number of updates (De Cao et al., 2021; Meng et al., 2022; Padmanabhan et al., 2023; Akyürek et al., 2024; Chen et al., 2023).

These studies suggest a natural parallel in the code setting: **can we efficiently update a pre-trained model's knowledge of an API?** In this work, we construct a benchmark to evaluate this capability. Our benchmark instances, shown in Figure 1, consist of a problem setting defined by a synthetic API update, such as an additional boolean flag in a function like numpy.argsort. We choose synthetic updates, as information about any real API function update will likely be used as a pre-training corpus by the next generation of pre-trained models. Then, for each function update, we have a number of program synthesis problems requiring the use of that update. Although there are solutions that do not use the update, the most parsimonious solutions do use the API functionality in question, and models are prompted to do so.

Our evaluation assesses whether LLMs can, after being updated on the synthetic API function update (docstring, example usage, etc.), solve these program synthesis examples using the given API function *without being provided the update at inference time*.

Our final benchmark, CodeUpdateArena, contains 670 program synthesis tasks, covering 54 functions from 7 Python packages. Our benchmark is synthetically constructed by a carefully designed data generation pipeline driven by GPT-4, enabling it to be scaled or updated with new instances in the future. We manually filter our generated API updates and conduct a number of additional intrinsic evaluations of dataset quality to establish the correctness of dataset instances.

Our experimental evaluation focuses on how existing small-scale LLMs (e.g., CodeLlama (Rozière et al., 2023)) perform at this update setting when combined with existing knowledge updating techniques. GPT-4 and Claude-3.5 are able to solve program synthesis examples when prompted with the API update in context, with Claude-3.5 outperforming GPT-4 on its own generated data. We then present two intuitive baselines for how practitioners would utilize API update information. The first method, fine-tuning models on a docstring explaining the update does not improve performance. However, fine-tuning on examples of the update being used does lead to improvement, and even outperforms having the API update in context. Through our ablation study, we found that the mix of training examples and learning rate are important for successful fine-tuning, but there is a tradeoff between efficacy and specificity of the update (impact on unrelated settings). We believe our dataset can provide a testbed for developing better methods for code knowledge editing in the future. Our code is released at ⭕.

# 2 BACKGROUND AND RELATED WORK

**Knowledge editing** Knowledge editing involves updating a pre-trained model's parameters to contain additional knowledge that was not present in its pre-training corpus. Suppose we have a model $\mathcal{M}$ and let $(c, u)$ denote the additional knowledge $u$ that should be returned in context $c$.

---

[1] https://youtu.be/outcGtbnMuQ?t=789

Past work has focused on finding a model $\mathcal{M}'$ such that $\mathcal{M}' \approx \mathcal{M}$ and $\mathcal{M}'(c)$ returns $u$ with high probability. For instance, suppose $c =$ *"the prime minister of the UK is"* and $u =$ *"Rishi Sunak"*; we want to update the model's knowledge about the UK's prime minister with as little change to other facts (e.g. *Eiffel tower is in Rome*) as possible.

Prior work quantifies model editing success by measuring whether $\mathcal{M}'$ can return $u$ when prompted with $c$. A second goal is to preserve the original $\mathcal{M}$ as closely as possible, measured by ensuring that the model's predictions on irrelevant contexts are not changed. The techniques for knowledge editing typically involve gradient updates (De Cao et al., 2021), including meta-learned updates (Mitchell et al., 2022), localized updates leveraging interpretability methods (Meng et al., 2022), and updates on a collection of related examples (Padmanabhan et al., 2023; Akyürek et al., 2024).

A third goal involves *knowledge propagation* (Onoe et al., 2023; Padmanabhan et al., 2023; Cohen et al., 2024; Powell et al., 2024; Zhong et al., 2023), where an LLM must be able to reason about the injected knowledge in contexts that may seem unrelated on the surface. However, current literature has many negative results for this setting (Cohen et al., 2024; Hua et al., 2024). Our benchmark will allow us to evaluate the state of affairs in the code setting, and whether functional competence around code updates is more easily obtained than functional competence around textual knowledge.

**Updates in Source Code**  Despite a large body of work on knowledge editing (Wang et al., 2023), past work in this space has not explored the ramifications for code language models. Rather than just reproduce an update like in knowledge editing settings (e.g. be able to generate *Python 3.12 has lifted restrictions on the usage of f-strings*), a user would likely expect a code LLM to be able to generate, debug, or otherwise reason about code containing these updates.

To the best of our knowledge, existing benchmarks mainly focus on general coding capabilities of LLMs rather than their capability in dealing with API updates or historical versions existent in pretraining corpus. Although some recent research has also explored providing documentations of functions (or tools)  (Zhou et al., 2022; Su et al., 2024; Zhang et al., 2023b; Hsieh et al., 2023) and code snippets (Su et al., 2024; Zhang et al., 2023a; Phan et al., 2024; Shrivastava et al., 2022) to LLMs in a retrieval-augmented framework (Chen et al., 2017; Guu et al., 2020; Lewis et al., 2020), our main focus is on enabling LLMs to internalize this knowledge in an update (in-weight) and propagate it during program synthesis as opposed to using it in-context. Therefore, our work also relates to more general program synthesis using LLMs (Austin et al., 2021), especially those on developing benchmarks (Chen et al., 2021; Liu et al., 2023; 2024; Gu et al., 2024; Jimenez et al., 2024; Ding et al., 2023; Du et al., 2023; Guo et al., 2024b; Xie et al., 2024; Lai et al., 2023).

**Defining an update taxonomy**  The goal of this work is to assess models' abilities to be updated with *realistic* changes to functions in APIs. Most of the time when new functionality is introduced, the update extends existing methods in an atomic way. For example, a new sorting algorithm is supported for argument `kind` in `numpy.argsort`. To systematically capture different types of updates, we create a taxonomy for function updates, capturing what operation (add/modify/delete) is used to update what component (function/argument/output) in what way.

## 3  TASK: CodeUpdateArena

We define $f \leftarrow u$ to be the update made to an existing function $f$ when providing it with new semantics $u$. Our task involves understanding whether a pretrained code language model $\mathcal{M}$ can be updated with $f \leftarrow u$. We assume that some kind of parametric update is made to yield a new model $\mathcal{M}^{(f \leftarrow u)}$; this can be done via various fine-tuning methods that have been proposed for knowledge editing. We will describe exactly how $u$ is conveyed to the language model in Section 6.1; here, we focus on what capabilities we want the updated model $\mathcal{M}^{(f \leftarrow u)}$ to exhibit.

To evaluate $\mathcal{M}^{(f \leftarrow u)}$, we provide a set of program synthesis examples $\mathcal{P}^{(f \leftarrow u)}$. Each program synthesis example consists of a problem scenario $s_i$, a problem specification $p_i$, and a set of $T$ unit test cases $\mathcal{T}_i^{(f \leftarrow u)} = \{(t_{i,1}, a_{i,1}), (t_{i,2}, a_{i,2}), \cdots (t_{i,T}, a_{i,T})\}$.

$$\mathcal{P}^{(f \leftarrow u)} := \left\{ \left( s_i, p_i, \mathcal{T}_i^{(f \leftarrow u)} \right) \right\}_{i=1}^{T}$$

Each example scenario and specification is related to the updated semantics $u$. Let $\tilde{c}_i \leftarrow \mathcal{M}^{(f \leftarrow u)}(s_i, p_i)$ denote the result of predicting a code solution to problem $i$ for update $u$. We want to evaluate $\mathcal{M}^{(f \leftarrow u)}$ for three broad capabilities: (1) **edit success**: $\forall j,\ \tilde{c}_i(t_{i,j}) = a_j$ (the update passes all test cases); (2) **use of** $f$**:** $\tilde{c}_i$ contains a call to the updated function $f$; (3) **specificity**: the update minimally changes the language model. See examples in Figure A.1.

Measuring whether samples from a code LLM pass test cases is typically done with pass@k (Chen et al., 2021). Drawing $k$ samples from an LLM, what is the probability that one of those samples passes the test cases? This can be computed analytically without bias by drawing $n > k$ samples, observing what number $c$ of those samples pass the test cases, and using the formula from Chen et al. (2021) (reproduced in Appendix D). In this work, we set $n = 5$ and $k \in \{1, 2, 5\}$.

**UPass@k**   Our main evaluation metric captures both **edit success** and **use of** $f$. We define UPass@k as the standard pass@k except that it only counts solutions that meaningfully use the updated function as "correct".

We run a solution against test cases with different function implementations at runtime:

   a) when executing *with the updated function* in the environment, the solution *must pass all* tests.

   b) when executing *with the old function* in the environment, the solution *must fail some* tests.

Details of how to do this execution are described in Appendix D. The first check is the standard one used in pass@k. This second check ensures that the new functionality of $f$ is leveraged in a nontrivial way. Detecting a call to $f$ is insufficient; if, for example, the update provides a new argument, we want the model to use that new argument rather than use $f$ in its pre-update form.

Our program synthesis examples are designed to be naturally suited to the updated function $f \leftarrow u$. It is, of course, possible for a code LLM to produce a solution that passes the tests but sidesteps the usage of $f$ altogether; however, in Section 6, we will see that prompted GPT-4 frequently *does* use the update in successful solutions.

**SPass@k**   captures how well the update is specific in that model's other capabilities are not affected (**specificity**) before ($\mathcal{M}$) and after ($\mathcal{M}^{(f \leftarrow u)}$) injecting each update. We discuss details in Section 6.1.

## 4   Update AND Arena GENERATION

We generate our data by prompting GPT-4 (Achiam et al., 2023) to instantiate our proposed task CodeUpdateArena, following recent work on generating synthetic datasets for complex tasks with LLMs (Sprague et al., 2024; Lee et al., 2024; Tang et al., 2024; Yehudai et al., 2024; Oh et al., 2024; Zhao et al., 2024). Each data instance requires an update semantics $u$ and program synthesis examples $\mathcal{P}^{(f \leftarrow u)}$ to evaluate the integration of the updates. We first generate the update semantics (described in Section 4.1) and generate program synthesis examples (Section 4.2). The output from each generation step is validated through manual inspection and heuristics. Figure 2 outlines our generation process.

### 4.1   Update (NEW API FUNCTION) GENERATION

**Step 1: Generate update specification** $u$   Given an update type (e.g., add new argument) and a function $f$ (e.g., numpy.argsort), we generate an update $u$ consisting of four pieces:

   • a *description* of the update: e.g., adding a new boolean argument 'reverse', which controls whether the sorting is descending or ascending.
   • the new function *signature*: e.g., numpy.argsort(..., reverse=False)
   • a *docstring* describing expected new behavior
   • the *rationale* behind this update

See Appendix B.5 for the details of the prompt. Notably, we generate the update providing the model only the function path and the function's docstring, obtained from the importlib library. See more details in Appendix B.1.

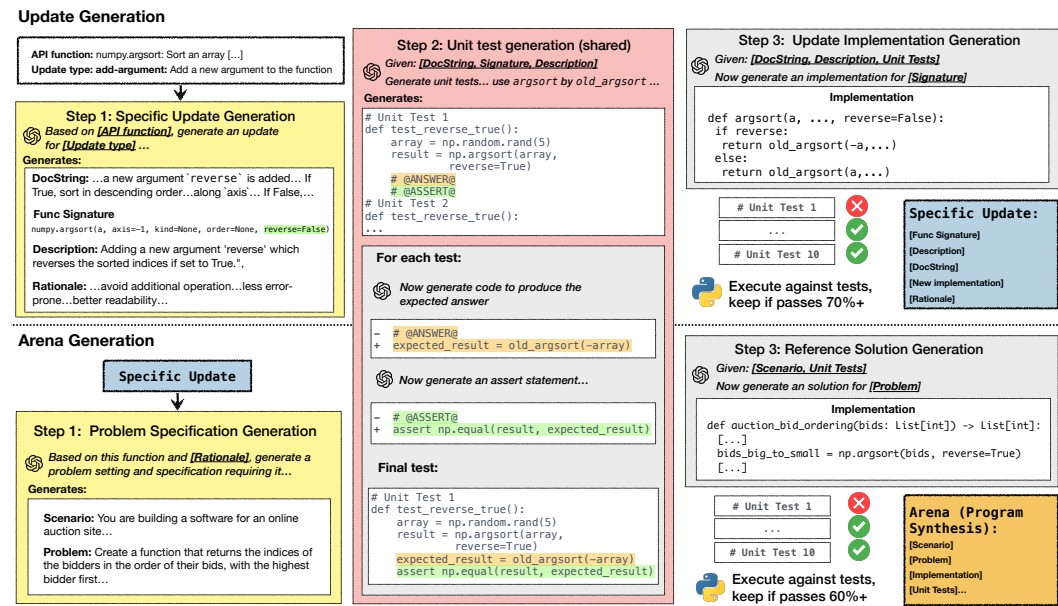

Figure 2: Overview of `CodeUpdateArena` generation pipeline. We first generate a spec for an update, unit tests for an update, and then the update's implementation. To generate program synthesis examples, we take an update, generate a problem specification, tests, and then a reference solution.

**Step 2: Generate a suite of unit tests**    Once the description of the update is available, we create a set of unit tests to verify the correctness of the updated function $f \leftarrow u$.

To make the tests comprehensive, we ask GPT-4 to generate 10 unit test functions, testing edge cases (e.g., empty input) and interaction with existing arguments (e.g. `reverse=True` and `axis=1`).

See Appendix B.5 for the details of the prompt. We first generate unit test "skeletons", unit test function with initialization of the input. Fig. 2 shows an example. Each skeleton takes the format of a unit test function with two placeholders — `@ANSWER@` for answer and `@ASSERT@` for assertion. Given a unit test skeleton, GPT-4 generates the answer and assertion statement(s). The details of answer and assertion generation can be found in Appendix B.2.

**Step 3: Generate an updated function $f \leftarrow u$**    We now prompt GPT4 to generate the source code for the updated function $f \leftarrow u$ given the function $f$ and update specification $u$. We prompt using the original function implementation (e.g. original `argsort`) to implement the new version. This typically involves an implementation that wraps the original version of the function; for instance, if a new boolean flag is added, call the function normally in one case and otherwise call it with a transformed input or output.

We validate the generated function with unit tests from the previous step. Specifically, we accept the updated function if (1) it passes 70% of unit tests and (2) it passes more unit tests than the original implementation.[2] To improve the coverage, we sample up to three implementations if earlier implementation does not satisfy two criteria above. After this process, on average, around 41% of update specifications are paired with an updated function implementation. The rest are discarded.

**Step 4: Filtering and deduplication**    Lastly, to verify the quality of generated data, the authors of this paper manually examine the update specifications and filter duplicates and trivial update specifications (e.g., change the return type from `list` to `tuple`). This process removes roughly 53% of examples on average, and the filtering percentage differs per package. We also filter update specifications for which we could not generate at least 3 valid program synthesis examples (37% of update specifications), as described in the next section.

---

[2]When a small number of unit tests are failed, they are often incorrect unit tests.

Table 1: Dataset size of `CodeUpdateArena` over seven python packages.

| Total # of unique functions | Total # updates | Total # PS examples | Total # unit tests in PS |
|:---:|:---:|:---:|:---:|
| 54 | 161 | 670 | 6.3 |

Table 2: The average number of tokens in generated update specs and program synthesis examples.

| Update: lengths in tokens | | | Program synthesis: lengths in tokens | | |
|:---:|:---:|:---:|:---:|:---:|:---:|
| description | docstring | function impl | scenario | problem specification | solution impl |
| 20.9 | 129.0 | 164.8 | 65.1 | 73.6 | 174.1 |

## 4.2 `Arena` (PROGRAM SYNTHESIS EXAMPLES) GENERATION

Having generated update semantics $u$ and the updated function implementation $f \leftarrow u$, we now generate program synthesis (PS) examples; see bottom half of Figure 2 and more details in Appendix B.6.

**Step 1: Problem specification** Given the update rationale generated as a part of update specification $u$, GPT4 generates: (1) a *scenario* $s_i$ that a problem is situated in; (2) the *problem specification* $p_i$ that a solution function is mean to fulfill; and (3) the solution's *function signature*, according to the problem specification. See an example at Appendix A.2.

**Step 2: Unit tests** We then generate a set of unit tests meant to test that the solution to the program synthesis example is correct. Note that these do not necessarily depend on the update, but only on the specification of the problem from Step 1; they do not test whether the function is used.

We allow GPT-4 to include updated function in its generation, in contrast with update generation, where GPT-4 could only call the old function through `old_[function name]`. Other than the difference above, the generation process is identical to Step 2 in update generation.

**Step 3: Reference Solution** The prompt instructs GPT-4 to solve the problem by using the new function as part of its solution. This helps to ensure that there exists a solution that uses the updated function. We define a threshold $\delta = 0.6$ of a fraction of tests that the implementation must pass in order to be included in the benchmark. We found this quality bar to be high enough given the presence of bad tests, which we discard next.

**Step 4: Filtering and Deduplication** Finally, we implement several filters of low-quality examples. First, we discard unit tests that generated solution doesn't pass, as well as unit tests checking for exceptions (`try/catch` behavior). Our inspection of these cases showed that failed unit tests are almost invariably incorrect while the generated reference solutions are correct. Second, for each update, we remove program synthesis cases for which reference solutions are too similar, to avoid GPT-4 generating essentially similar solutions. Example of duplicate reference function in Figure 9. See more detailed description at Appendix B.3

**Step 5: Literalize answers in unit tests** During generation, many unit tests initially rely on calling updated APIs themselves to produce the correct answer to the test programmatically. However, this causes unintended failures in the unit tests when running the old API updates in the environment, leading to false positives for `UPass@k` even when the synthesis code does not use the new function. We "literalize" the unit tests to remove these usages of the API; we provide more details in Appendix B.4.

## 5 CHARACTERIZING THE DATASET

Table 1 gives the statistics of our updates (161) and Table 2 gives the statistics of the final arena program synthesis examples (670). Each update features at least three program synthesis examples. Figure 7 in the Appendix shows the distribution of how many program synthesis examples are included per update. We give further statistics about the diversity of function updates in the Appendix: Figures 5 and 6 and Table 8 to show the diversity of packages and update types that our function updates reflect. Finally, Figure 8 shows the average edit distances between solutions to our program

Table 3: GPT-4's `pass@5` score on our benchmark and the number of instances per package

| Package | itertools | math | numpy | pandas | re | sympy | torch | Avg. |
|---------|-----------|------|-------|--------|------|-------|-------|------|
| pass@5 | 75.6 | 89.0 | 85.8 | 87.1 | 75.8 | 91.7 | 86.8 | 85.1 |
| count | 45 | 182 | 141 | 93 | 91 | 12 | 106 | − |

synthesis examples. Despite using the same prompt, we see that the sampled solutions to different examples differ substantially. A full example from our dataset can be found in Appendix A.2.

**Solvability**    We also demonstrate that our program synthesis examples are solvable: do the problem scenario and specification provide enough detail to actually synthesize the correct code? To test this, we run an experiment prepending the update docstring to GPT-4's context and evaluating `pass@k` *without* checking for whether the update was correctly used. As shown in Table 3, GPT-4 achieves `pass@5` of 85.1; this means, in most scenarios, GPT-4 is able to provide *a* correct solution to the program synthesis examples within 5 trials. The performance is reasonably high across all packages in the benchmark.

**Human Inspection**    We conducted manual inspection on predicted solution that GPT-4 fails in Table 3, and categorized sources of error. See details in Appendix C.2. We found errors mostly come from "Wrong Solution" and "Incomplete Solution", meaning failure to handle the edge cases of the problem statement, real mistakes due to misinterpretation of the problem statement, etc. These errors can be avoided by using stronger language models, as we will demonstrate in Section 6.2. We observed relatively few cases of incorrect test cases or bad specifications, indicating that our dataset is of sufficient quality to test knowledge editing methods. We also verify the quality of our generated data by measuring the unit test coverage on our reference solution in Appendix C.3.

# 6 EXPERIMENTS

## 6.1 EXPERIMENTAL SETTING

**Base LLMs**    We tested two proprietary models for our prepending experiment: GPT-4 (`gpt-4-0613`) (Achiam et al., 2023) and Claude-3.5 (`claude-3-5-sonnet-20240620`) (Anthropic, 2024). For fine-tuning experiments, we consider three open-source code LLMs that are instruction-tuned: CodeLlama (7B-sized; Rozière et al. (2023)), DS-Coder-v1 (6.7B), and DS-Coder-v1.5 (7B; Guo et al. (2024a)).

**Evaluation Scenario**    We evaluate approaches in the single-edit scenario, where we inject one update at a time about a single API. For measuring efficacy (`UPass`), we consider whether the predicted solution passes all the unit tests with the updated API but fails to do so with the old API (see Section 3 and Appendix D). To measure specificity (`SPass`), we measure the change in model performance on a random sample of 82 HumanEval (Chen et al., 2021) instances across 25 random single edits.

**Knowledge Editing Approaches**

- **Prepend**    In this setting, we simply prepend the function update's docstring in-context at inference time (see Prompt E.3). This represents a retrieval-augmented (RAG) setting (Su et al., 2024; Zhou et al., 2022), which leads to higher inference cost and does *not* represent model updates. This is not considered a knowledge editing approach but establishes the performance of an effective alternative method (Onoe et al., 2023; Padmanabhan et al., 2023).
- **Fine-tune on update information: FT (U)**    In this setting, we conduct continued pretraining on the docstring describing the new behavior (Gururangan et al., 2020). This setting captures the scenario where the package designer provides a release note about the updated API function while no examples of the function being used are available.
- **Fine-tune on program synthesis examples: FT (PS)**    In this setting, we conduct supervised finetuning on the program synthesis examples, informing LLM how the new functions should be used. Such program synthesis examples can be collected from the API documentation, cutting-edge repositories, or generated to update code LLMs. To implement this, we select $N_u$ examples

Table 4: Knowledge editing results on `CodeUpdateArena`. *: comparing against the base model, the gap is significant according to a paired bootstrap test with $p < 0.05$.

| Base Model | Approach | UPass (Efficacy) ↑ | | SPass (Specificity) ↑ | | Pass with updated API ↑ | |
|---|---|---|---|---|---|---|---|
| | | @1 ($\Delta$) | @5 ($\Delta$) | @1 ($\Delta$) | @5 ($\Delta$) | @1 ($\Delta$) | @5 ($\Delta$) |
| GPT-4 | Base Model | 2.7 | 5.7 | – | – | 54.1 | 74.5 |
| | Prepend | $34.1^*_{+31.4}$ | $57.0^*_{+51.3}$ | – | – | $63.9^*_{+9.7}$ | $83.0^*_{+8.5}$ |
| Claude-3.5 | Base Model | 2.9 | 3.6 | – | – | 51.8 | 61.0 |
| | Prepend | $58.7^*_{+55.9}$ | $71.9^*_{+68.4}$ | – | – | $68.4^*_{+16.6}$ | $77.2^*_{+16.1}$ |
| CODELLAMA | Base Model | 4.4 | 7.6 | 39.8 | 50.0 | 28.4 | 39.4 |
| | Prepend | $6.7^*_{+2.4}$ | $10.6^*_{+3.0}$ | – | – | $32.0^*_{+3.6}$ | $44.6^*_{+5.2}$ |
| | FT (U) | $4.3_{\ -0.1}$ | $7.3_{\ -0.3}$ | $28.8^*_{-10.9}$ | $45.9^*_{-4.1}$ | $28.0_{\ -0.4}$ | $40.9_{\ +1.5}$ |
| | FT (PS) | $22.9^*_{+18.6}$ | $37.6^*_{+30.0}$ | $17.0^*_{-22.8}$ | $37.1^*_{-12.9}$ | $28.6_{\ +0.1}$ | $45.7^*_{+6.3}$ |
| DS-CODER-v1 | Base Model | 2.9 | 5.2 | 49.3 | 79.3 | 30.3 | 46.6 |
| | Prepend | $10.3^*_{+7.5}$ | $19.6^*_{+14.3}$ | – | – | $35.1^*_{+4.8}$ | $53.4^*_{+6.9}$ |
| | FT (U) | $3.1_{\ +0.3}$ | $6.1_{\ +0.9}$ | $40.0^*_{-9.2}$ | $74.0^*_{-5.2}$ | $33.5^*_{+3.2}$ | $51.6^*_{+5.1}$ |
| | FT (PS) | $27.7^*_{+24.8}$ | $44.0^*_{+38.8}$ | $52.5^*_{+3.3}$ | $78.4_{\ -0.8}$ | $38.3^*_{+7.9}$ | $58.7^*_{+12.1}$ |
| DS-CODER-v1.5 | Base Model | 3.2 | 6.4 | 67.1 | 79.3 | 46.8 | 64.3 |
| | Prepend | $11.8^*_{+8.6}$ | $22.1^*_{+15.7}$ | – | – | $50.9^*_{+4.1}$ | $70.7^*_{+6.4}$ |
| | FT (U) | $3.6_{\ +0.4}$ | $7.0_{\ +0.6}$ | $56.4^*_{-10.7}$ | $77.3^*_{-2.0}$ | $47.0_{\ +0.2}$ | $65.4_{\ +1.0}$ |
| | FT (PS) | $29.4^*_{+26.2}$ | $47.2^*_{+40.7}$ | $37.3^*_{-29.8}$ | $61.2^*_{-18.0}$ | $38.7^*_{-8.1}$ | $61.3_{\ -3.0}$ |

demonstrating the target update and repeat them $c$ times, combined with $N_r$ examples from $r$ random updates in the rest of our dataset. We found adding such random examples improves the performance, potentially because it helps the model learn the update style and retain information about existing functions. In this work, $N_u$ is fixed to be 2 because many updates only have 3 examples; we adopt a cross-validation scheme for evaluation. See more description on evaluation in Appendix E.3. We provide detailed ablation for our design choices in Section 6.3.

**Training Details**   We use LoRA for all finetuning experiments (Hu et al., 2022). We choose our learning rate of 1e-3 from 1e-8 to 1e-2 on a subset of our data to balance `UPass` and `SPass`. See more details for experiments configuration and prompt for training and testing in Appendix E.

**Non-applicability of existing knowledge editing methods**   Although a number of methods for knowledge editing have been proposed, not all of them are applicable to our setting. A line of methods including ROME, MEMIT, and REMEDI (Meng et al., 2022; 2023; Hernandez et al., 2023) assume the injected data follows a strict knowledge triplet format of (`subject, relation, object`); this triplet structure is required for localization. Applying those methods to `CodeUpdateArena` is not straightforward, as code entities do not exhibit these knowledge graph-like relations. Other methods designed for similar settings do not assume this structure. However, even these more flexible approaches like MEND (Mitchell et al., 2022) and others (Hartvigsen et al., 2023; Huang et al., 2023) are optimized for models regurgitating the right short phrase response, typically less than 10 tokens. Our reference solutions contain 175.3 tokens on average. Furthermore, these methods have not proven effective in more related natural language settings such as Onoe et al. (2023).

## 6.2   RESULTS AND DISCUSSIONS

We present the experimental results in Table 4. All of the open-source models perform worse than proprietary models in the Prepend setting. GPT-4 and Claude-3.5 both achieve high performance. Interestingly, Claude-3.5 outperforms GPT-4 despite GPT-4 having been used to generate the dataset; this suggests that GPT-4 is not strongly favored on this benchmark beyond other frontier models.

Similar to the results in entity knowledge editing (Onoe et al., 2023), continuing training on update information (FT (U)) does not improve efficacy and hurts specificity. On the other hand, training the model on program synthesis examples (FT (PS)) works well, outperforming the prepend setting (See Table 4). We observe that, except on DS-Coder-v1, the open models suffer a large drop in performance on specificity. This means that although our method can use the updated API to solve downstream

Table 5: Experiments with different training set construct controlled by $(c, r)$. Our standard setting is $(2, 1)$. We see that the update itself is required to do well; training on unrelated examples is much worse (compare $(0, 3)$). However, including random update(s) in training data is beneficial when paired with the update (compare $(3, 0)$). *: comparing against base model, the gap is significant according to a paired bootstrap test with $p < 0.05$. See additional results in Table 11.

| Method | $c$ | $r$ | UPass (Efficacy) ↑ | | SPass (Specificity) ↑ | |
| --- | --- | --- | --- | --- | --- | --- |
| | | | @1 ($\Delta$) | @5 ($\Delta$) | @1 ($\Delta$) | @5 ($\Delta$) |
| DS-Coder-v1.5 | — | — | 2.6 | 4.3 | 67.1 | 79.3 |
| + Prepend | — | — | $12.4^*_{+8.8}$ | $21.7^*_{+14.3}$ | — | — |
| + FT (PS) | 2 | 1 | $34.8^*_{+31.2}$ | $50.3^*_{+42.9}$ | $37.3^*_{-29.8}$ | $61.2^*_{-18.0}$ |
| | 1 | 2 | $28.1^*_{+25.5}$ | $45.3^*_{+41.0}$ | $32.3^*_{-34.8}$ | $63.8^*_{-15.5}$ |
| | 3 | 0 | $31.9^*_{+28.3}$ | $44.7^*_{+37.3}$ | $40.2^*_{-26.9}$ | $61.9^*_{-17.4}$ |
| | 0 | 3 | $9.6^*_{+6.0}$ | $21.1^*_{+13.7}$ | $35.7^*_{-31.4}$ | $66.3^*_{-13.0}$ |

tasks better than other baselines, retaining performance and avoiding catastrophic forgetting remains a key challenge.

Our results echo with prior work in that training models to regurgitate the injected knowledge does not help models to pragmatically use the knowledge for downstream tasks (Zhong et al., 2023) or update the status of related knowledge (Cohen et al., 2024; Jiang et al., 2024; Allen-Zhu & Li, 2024). In contrast, providing models with a more direct training signal like in our FT (PS) baseline or related context (Padmanabhan et al., 2023; Akyürek et al., 2024) helps with knowledge propagation and utilizing the knowledge pragmatically.

In the following section, we will conduct an ablation study on our methods to understand the importance of different design choices of our methods across different injected models.

### 6.3 ABLATION STUDY

Our method FT (PS) serves as a starting point for future editing methods on our dataset to build on. In this section, we study its design choices and demonstrate the difficulties and the trade-offs to achieve efficacy and specificity at the same time. For efficiency, we follow the same evaluation procedure as Table 4 and but only use 1 random program synthesis example per update (in total, 161) to calculate efficacy. We focus on the DS-Coder-v1, which achieves the best specificity and overall performance, and DS-Coder-v1.5, which achieves the highest efficacy in finetuning experiments.

**Impact of training data for FT (PS)** In our main experiment, the training set consists of $c = 2$ copies of $N_u$ examples from target update and $N_r = 2$ examples from $r = 1$ random updates.[3] In this section, we investigated how the construct of training data affects knowledge injection, by changing the values of $c$ and $r$ while fixing $c + r$.

Having the target update is important: when we train the model on only program synthesis examples from random updates, Tables 5 and 11 show little or negative gain from learning only the task format. However, the random synthesis examples also matter: when we exclude them, the performance decreases as well, although to a less extent (see Table 11).

In a more complete hyperparameter sweep, we found that repeating the examples from target update twice ($c = 2$) is generally the optimal hyperparameter, beyond which we observe diminishing gains in efficacy and drops in specificity. Second, although different models have different optimal values, we found that larger number of random updates $r$ will continue to decrease models' performances for efficacy and specificity. This is a different observation from prior work (Gangadhar & Stratos, 2024). See more details in Appendix E.4.

**Different models have different sensitivity to learning rates** Efficacy and specificity often show tradeoffs. We vary the learning rate and plot these in Figure 3 to better understand this relationship. We observe that DS-Coder-v1 and DS-Coder-v1.5 have different sensitivities to the learning rate.

---

[3] We take a pair of unique program synthesis examples from each $r$ random updates

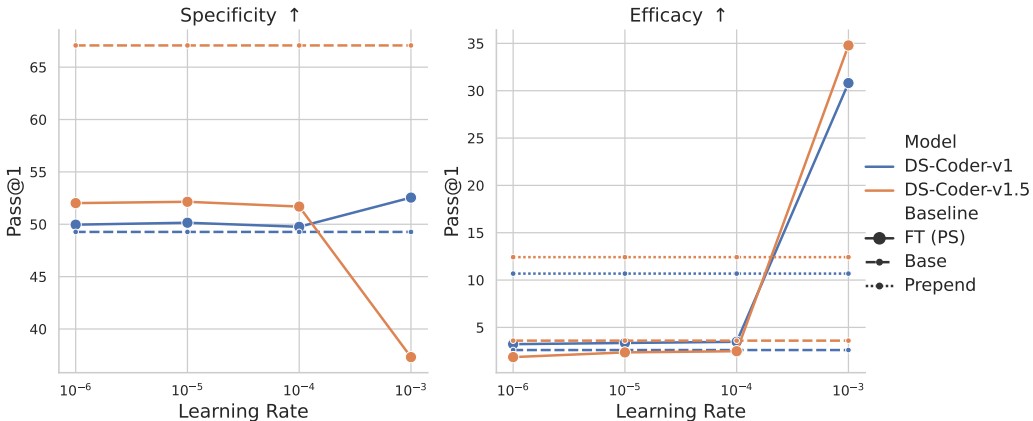

Figure 3: Sensitivity test on learning rate. Sensitivity for specificity is model-specific and may have trade-offs with efficacy. A large enough learning rate (e.g. 1e-3) is required to outperform the prepend setting.

First, knowledge injection, even with a learning rate as small as 1e-6, greatly harms DS-Coder-v1.5's performance on HumanEval whereas DS-Coder-v1's performance on HumanEval is kept unharmed. Secondly, both models start to outperform the prepend setting with a learning rate greater than 1e-4. Furthermore, as the learning rate increases, the specificity and efficacy of DS-Coder-v1.5 exhibit a clear tradeoff — when the learning rate increases beyond 1e-4, DS-Coder-v1.5 undergoes a large increase in efficacy and decrease in specificity. In contrast, DS-Coder-v1's efficacy increases even with an improvement in specificity. We verify the gap from the Base model by a paired bootstrap test with $p < 0.05$. However, we do not observe this increase in both metrics in general (e.g., Figure 12). We believe future work needs to investigate the cause of such differences and take them into account when designing new algorithms.

## 7 CONCLUSION

In this paper, we presented `CodeUpdateArena`, a benchmark of API updates and corresponding program synthesis examples. We demonstrated that our approach to synthesizing these leads to high-quality examples. Across three LLMs, we conduct experiments for two simple baselines. One of the baselines greatly outperforms prepending update information in context, which is different from observation from knowledge editing in entity-driven scenarios. We further conducted a comprehensive ablation study to inform future exploration. We hope our initial exploration could spur future work to develop new knowledge updating methods for code LLMs to benchmark on this setting.

**Limitations:** One limitation of `CodeUpdateArena` is that certain APIs are difficult to test with our dataset synthesis framework. For instance, it is difficult to generate unit tests for machine learning APIs, and can be very involved to generate tests if a significant setup is needed (e.g., a mock web server backend). Furthermore, our focus on synthetic API updates is necessary to avoid data contamination, but at the same time decreases the realism of our dataset. It would be ideal to have real software engineers annotate these kinds of updates at scale, but in preliminary experiments, we found it very difficult to come up with creative and realistic updates. Finally, our examples are restricted to Python and English-language descriptions; we believe a multilingual version of the benchmark (both human languages and code languages) would be useful.

**Reproducibility Statement:** Our dataset and code are available at ⬛. The dataset construction procedure is fully automated and documented, enabling future researchers to generate similar or related datasets. For our experiments, we detailed our hyperparameters in Appendix E. Our ablation study (Section 6.3 and Appendix E.4) depicts the model's behavior across different design choices.

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

# A DATASET

## A.1 UPDATE TAXONOMY

To systematically capture different types of updates, we first create a taxonomy for update types, rooted in updates to functions.

Recall that we define $f \leftarrow u$ to be the update made to an existing function $f$ when providing it with new semantics $u$. We assume $f$ always takes the form of Function([argument1, argument2, $\cdots$]) $\rightarrow$ Output. We view $u$ as consisting of three independent components: (1) the Action that the update is applying to the API function (e.g., deprecate); (2) the Locus that the action happens at (e.g. argument or output); (3) and the Aspect that the action is applying at some place for (e.g., name and data type). See Table 6 in the appendix for possible values of each component. We note that we do not focus on Action=deprecate in this work, as techniques for knowledge unlearning are different than those for knowledge editing.

An *update type* is a tuple with values for each component listed in Table 6. For example, the add-argument-NULL update type means the update is adding a completely new argument to the existing arguments of a function, and modify-argument-name update type means the update is modifying the name of an existing argument (i.e. renaming). We note that not all combination makes sense, e.g. modify-output-name; or some update types might overlap with another e.g., add-function-semantics overlaps with modify-function-semantics. We remove those and obtain 17 update types.

Table 6: Update Taxonomy components

| Component | Values |
|---|---|
| Action | {add, modify, deprecate} |
| Locus | {function, argument, output} |
| Aspect | {NULL, name, data_type, default_value, supported_value} |

## A.2 EXAMPLE

We present a complete example from our dataset below. The unit tests for the update itself are omitted as these are not used by any of our methods and are only used for quality control.

---

**A.1 Example Data**

**Update Description:**
A new boolean parameter 'inverse' is added to math.pow() to calculate the inverse power.

**Update DocString:**
An additional parameter 'inverse' has been introduced to the function signature, which when set to True, will return the inverse power of the numbers, i.e., 1/(x^y). This results in a non-trivially different implementation from the previous one, and the rest of the function behavior stays the same. The new parameter 'inverse' is a boolean parameter with a default value of False. When 'inverse' is set to True, the output of the function is changed to 1/(x^y), and when 'inverse' is set to False or left unspecified, the output remains the same as in the old version, which is x^y.

**Rationale:**
Sometimes users might need to calculate the inverse power (1 to the power of y divided by x) and this feature saves them from having to manually calculate the inverse power of a number.

**Program:**
# "problem": Alan needs to compute present values of these future cash flows for 'n' periods and 'r' different rates. However, computing it manually or using traditional Python methods is cumbersome and prone to

---

errors. Help Alan by creating a program that can compute this efficiently for any 'n' and 'r'.

**Scenario:**
Alan is a property investor who has recently invested in commercial projects, where the rental income fluctuates. He came across an investment formula $(1/(1 + r)^n)$ that can approximate the present value of future cash flows. Here, 'r' represents the discount rate or interest rate, and 'n' represents the number of cash flow periods.

**Solution Signature:**
def compute_present_value(r: float, n: int) -> float:

**Updated API:**

```python
# Import the necessary library.
import math
def compute_present_value(r: float, n: int) -> float:
    # Check for invalid inputs.
    if r < 0:
            raise ValueError("Rate cannot be negative.")
    if n < 0:
            raise ValueError("Number of periods cannot be negative.")

    # Use the updated math.pow() API to calculate the present value.
    return math.pow(1.0 + r, n, inverse=True)
```

**Unit Tests:**

```python
# Unit test 0
def test_compute_present_value_small_inputs():
    r = 0.1
    n = 3
    # small inputs for rate and number of periods
    result = compute_present_value(r, n)
    import math
    expected_result = math.pow(1 + r, n, inverse=True)

    # Check equivalence between 'result' and 'expected_result'
    assert result == expected_result
# Unit test 1
def test_compute_present_value_large_inputs():
    r = 0.9
    n = 100
    # large inputs for rate and number of periods
    result = compute_present_value(r, n)
    import math

    # Since the inverse is required, we set 'inverse' to True in math.pow()
    expected_result = math.pow(1 + r, n, inverse=True)

    assert result == expected_result, f"Expected {expected_result} but got {result}"
# Unit test 2
def test_compute_present_value_zero_rate():
    r = 0.0
    n = 10
    # testing with 0 rate should compute to the cash flow amount
    result = compute_present_value(r, n)
    expected_result = 1.0
```

```python
    assert result == expected_result, f"Expected {expected_result}, but got {result}"
# Unit test 3
def test_compute_present_value_zero_periods():
    r = 0.5
    n = 0
    # testing with 0 periods should compute to the cash flow amount
    result = compute_present_value(r, n)
    expected_result = math.pow((1 + r), -n, inverse=True)

    assert result == expected_result,
    ↪  f"Error: Expected result {expected_result}, but got {result}."
# Unit test 4
def test_compute_present_value_negative_rate():
    try:
            r = -0.1
            n = 5
            # negative rate should raise an exception
            compute_present_value(r, n)
    except Exception:
            assert True
    else:
            assert False
# Unit test 5
def test_compute_present_value_negative_periods():
    try:
            r = 0.1
            n = -5
            # negative number of periods should raise an exception
            compute_present_value(r, n)
    except Exception:
            assert True
    else:
            assert False
# Unit test d
def test_compute_present_value_large_rate():
    r = 1.5
    n = 10
    # large rate should lead to small present value
    result = compute_present_value(r, n)
    from math import pow

    expected_result = pow(1 + r, n, inverse=True)

    assert abs(result - expected_result) <= 1e-9,
    ↪  f"Expected {expected_result}, but got {result}"
# Unit test 7
def test_compute_present_value_one_period():
    r = 0.2
    n = 1
    # one cash flow period should return a simple discounted value
    result = compute_present_value(r, n)
    expected_result = math.pow(1 + r, n, inverse=True)

  assert result == expected_result, f"Expected {expected_result}, but got {result}"
# Unit test 8
def test_compute_present_value_many_periods():
    r = 0.1
    n = 30
    # more periods should accumulate more discount
    result = compute_present_value(r, n)
    import math
```

```
    # compute the expected_result using the provided formula 1/(1 +
    ↪  r)\textasciicircum n
    expected_result = math.pow(1 + r, n, inverse=True)

    # At this point, we are checking the equivalence between `result` and
    ↪  `expected_result`
    assert result == expected_result, f'Expected {expected_result}, but got {result}'
# Unit test 9
def test_compute_present_value_edge_rate():
    r = 1.0
    n = 10
    # edge rate of 1.0 should also be handled
    result = compute_present_value(r, n)
    import math
    expected_result = math.pow(1 + r, n, inverse=True)

    assert result == expected_result, f"Expected {expected_result}, but got {result}"
```

# B  DATA GENERATION DETAILS

## B.1  PREPROCESSING API PATH

As discussed in Section 4.1, most of the time, we are able to retrieve full information about a function using the `importlib` and `inspect` packages. For our implementation, we decided to also separately extract a function's argument. However, these sometimes might not be possible using `importlib` and `inspect` package. Therefore, we devise two fallback options: (1) use regular expression to extract them from documentation; and if that fails, (2) we feed the docstring to GPT-4 and have it write arguments for us. We include the prompt for doing so in Appendix B.5.

## B.2  UNIT TEST GENERATION

For answer generation (`@ANSWER@`), we let GPT-4 choose between the following strategies:

1. directly write out the literal values of the answer (e.g. `numpy.array([1, 0, 2]`);

2. *or* write a step-by-step code snippet Wei et al. (2022) to accomplish the calculation in which it could call the old API function through `old_argsort(array[::-1],...)` (e.g. random input in Fig. 4).

For assertion generation (`@ANSWER@`), we note that objects in different packages require different ways to check equality, for example, instead of "==", one needs to use `numpy.equal` for `numpy.array`; and `df.equals` for `pandas.DataFrame`. To make sure the assertions are appropriately generated, we use package-specific prompts to guide GPT-4 generation. See our package instructions at Prompt B.9.

Figure 4: Example of unit test skeleton

```
def test_reverse_true():
    array = np.random.rand(5), ax = -1
    result = np.argsort(array, axis=ax,
    ↪  reverse=True)
    # @ANSWER@
    # @ASSERT@
```

## B.3  DEDUPLICATION

To deduplicate program synthesis examples, we first canonicalize each reference solution for function and variable names. Then, we compare the edit distances among Program Synthesis examples' reference solutions per update. We discard one of the examples in each pair with an edit distance of less than 25. If after discarding, # PS is equal to 1, we will keep both program synthesis examples. The mean edit distance for those "allowed duplicates" is $17.0 \pm 7.1$. In total, we remove 134 examples.

Table 7: Knowledge editing results on `CodeUpdateArena` *without* literalizing unit tests. *: comparing against the base model, the gap is significant according to a paired bootstrap test with $p < 0.05$.

| Base Model | Approach | UPass (Efficacy) ↑ | | SPass (Specificity) ↑ | | Pass with updated API ↑ | |
|---|---|---|---|---|---|---|---|
| | | @1 ($\Delta$) | @5 ($\Delta$) | @1 ($\Delta$) | @5 ($\Delta$) | @1 ($\Delta$) | @5 ($\Delta$) |
| GPT-4 | Base Model | 22.7 | 33.3 | – | – | 54.0 | 74.6 |
| | Prepend | $42.7^*_{+20.0}$ | $63.6^*_{+30.3}$ | – | – | $64.5^*_{+10.4}$ | $84.0^*_{+9.4}$ |
| Claude-3.5 | Base Model | 21.8 | 26.7 | – | – | 52.2 | 61.6 |
| | Prepend | $61.5^*_{+39.7}$ | $73.4^*_{+46.7}$ | – | – | $69.5^*_{+17.3}$ | $78.4^*_{+16.7}$ |
| CODELLAMA | Base Model | 12.2 | 17.5 | 39.8 | 50.0 | 28.5 | 39.9 |
| | Prepend | $14.7^*_{+2.5}$ | $21.0^*_{+3.5}$ | – | – | $32.2^*_{+3.7}$ | $45.4^*_{+5.5}$ |
| | FT (U) | $11.3_{-0.9}$ | $17.6_{+0.1}$ | $28.8^*_{-10.9}$ | $45.9^*_{-4.1}$ | $26.5^*_{-1.9}$ | $39.6_{-0.3}$ |
| | FT (PS) | $24.2^*_{+12.0}$ | $39.4^*_{+21.9}$ | $17.0^*_{-22.8}$ | $37.1^*_{-12.9}$ | $28.2_{-0.3}$ | $45.1^*_{+5.2}$ |
| DS-CODER-V1 | Base Model | 12.2 | 20.4 | 49.3 | 79.3 | 31.0 | 48.2 |
| | Prepend | $18.1^*_{+5.9}$ | $29.4^*_{+9.0}$ | – | – | $35.3^*_{+4.3}$ | $53.6^*_{+5.4}$ |
| | FT (U) | $12.7_{+0.5}$ | $20.9_{+0.5}$ | $40.0^*_{-9.2}$ | $74.0^*_{-5.2}$ | $32.7^*_{+1.8}$ | $50.4_{+2.2}$ |
| | FT (PS) | $30.6^*_{+18.4}$ | $47.6^*_{+27.2}$ | $52.5^*_{+3.3}$ | $78.4_{-0.8}$ | $37.9^*_{+6.9}$ | $58.1^*_{+9.9}$ |
| DS-CODER-V1.5 | Base Model | 20.0 | 29.4 | 67.1 | 79.3 | 46.8 | 64.8 |
| | Prepend | $25.8^*_{+5.8}$ | $38.4^*_{+9.0}$ | – | – | $51.3^*_{+4.5}$ | $71.5^*_{+6.7}$ |
| | FT (U) | $20.1_{+0.1}$ | $29.9_{+0.5}$ | $56.4^*_{-10.7}$ | $77.3^*_{-2.0}$ | $47.0_{+0.2}$ | $66.1_{+1.3}$ |
| | FT (PS) | $32.7^*_{+12.7}$ | $52.1^*_{+22.7}$ | $37.3^*_{-29.8}$ | $61.2^*_{-18.0}$ | $38.2^*_{-8.6}$ | $60.3^*_{-4.5}$ |

## B.4   LITERALIZE ANSWER IN UNIT TEST

We define *literalizing* a unit test as taking the unit test, which may call an updated API, and turning it into a semantically equivalent version that does not call the updated API. To do this, we obtain answers from unit tests (e.g., `pickle.dump(...)`), turn the Python object to literal string, e.g. `numpy.array2string()`, and replace the original answer section with a simple assignment `expected_result = ....`. This can be challenging in a few cases:

1. when the input of the unit tests is randomly initialized (e.g., `torch.randn`);

2. when the input is initialized with a large dimension (e.g., `image = numpy.full((1000, 1000)))` and result in very large literal values;

3. when an object like `pandas.Dataframe` might contain metadata that is hard to generally capture during literalization, or requires changes in the assertions section like `re.Match` object.

We use `pickle` serialization and deserialization to literalize tests, and when this process fails as in the cases above, we invoke Claude-3.5-sonnet to edit the unit test to make the appropriate changes while preserving the semantics. After processing, we turn the answers of 4114 unit tests into literal values (out of 4221 unit tests).[4] In Table 7, we include the evaluated results without literalization, which shows models like Base model has substantially higher `UPass@k`.

## B.5   GENERATION PROMPT: UPDATE

See Prompt B.1 for docstring summarization.

See Prompt B.2 for inferring the function arguments from the function path, e.g. `numpy.argsort`

See Prompt B.3 for generating update specifications

See Prompt B.4 for generating unit test skeletons

See Prompt B.5 for generating unit test answers; part of the prompt takes corresponding instruction from Prompt B.9 to guide the model to generate for different packages.

See Prompt B.6 for generating unit test assertions; part of the the prompt takes corresponding instruction from Prompt B.9 to guide the model to generate for different packages.

---

[4]We verify the correctness by executing reference solution on the units and receiving perfect performance.

See Prompt B.7 for generating function update implementation

See Prompt B.8 for generating missing imports given any code.

See Prompt B.9 for different packages when generating assertions and answers.

### B.1 Update: Docstring summarization

**System prompt:**
You are a helpful assistant.
You will be given documentation for an API in a popular Python library.

You need to do the following:
1: You MUST extract descriptions about the functionality, input parameters, and output from the original documentation.
2: You could include some illustrative code in the summary if the summary is ambiguous.
3: You MUST keep the most important information, e.g. description, data type, etc.
4: The reader of your summary MUST be able to implement the function with summarized documentation.
5: You MUST maintain the original structure, format, and the style of the documentation.
6: Output the summarized documentation in text.

**User Prompt:**
{{docstring, e.g. numpy.argsort.__doc__}}

### B.2 Update: Prompt to Infer Argument

{: System prompt}
Infer argument of a Python function signature from documentation (output of `[full_api_path].__doc__`).

Function signature takes the form of
```
[full_api_path]([arguments])
```
Output the right [arguments].

Note:
* Output raw text.
* DO NOT Wrap output in a Python code block.
* DO NOT include documentation in the output.

{: User Prompt}
Full API path:
{{{{full_api_path}}}}

Documentation:
{{{{documentation}}}}

### B.3 Update: Update Specification

**System prompt:**
You are a helpful assistant. You think deeply and creatively.
Your task is to assist users to think of and instantiate interesting cases of API update.

A desirable update should satisfy the following criteria:
* The update should make the call site of the old function to be un-executable and one need to follow the new function signature.
* The update should be as atomic as possible. It only includes one of the three possible editing actions and only happens to one place of the functions. So that the new function signature and old signature only differs

at one place.
* The update should lead to a new function signature whose implementation is non-trivially different from the old ones. An undesirable result is that the new implementation trivially calls the old function.
* The update should be a sensible change that fits the overall topic of the function and the Python library.
* The update should NOT contradict existing functionality of the old function.
* The update needs to be supported by a good reason for library designer to introduce it

Return the entire response in JSON format as a dictionary. Make sure nested brackets are closed correctly. Be careful with unterminated string literal. The dictionary should contain the following:

1: "update_description": (as string) a short one-sentence description of the update.
2: "rationale": (as string) why any hypothetical designer of the API might want to introduce these changes.
3: "new_function_signature": (as string) the new function signature.
3.1: "new_function_signature" MUST start with the full reference to the function. For example, "numpy.mean" instead of "def mean".
4: "update_docstring": (as string) the added documentation that explains the new functionality of the atomic update. It MUST be self-contained, unambiguous, detailed but concise.
4.1: You MUST succinctly explain the updated behavior of the new API, and how it differs from the old behavior.
4.2: The "update_docstring" MUST fully specify the behavior about the update. For example, how the changes in input would change the output of new API w.r.t. the old version.
4.3: A third-person MUST be able to develop a new implementation by just reading the "update_docstring" along with the old docstring.
4.4: "update_docstring" could take the form of natural language, numpy-style docstring, pseudo-code examples, etc. Make the most sensible choice. If it's a string with multiple lines, output "
n" as line break.
4.5: DO NOT include example(s) of using the updated API in "update_docstring".

You will be given a function signature, optionally along with its docstring, and the Python library it belongs to. You will think what realistic update could happen to the function signature.

Give me 1 example of possible update(s) that a new function argument is added.

**User Prompt:**
Package: {{parent_path}}

[DOC]
def {{function_signature}}
{{summarized_doc_string}}
[/DOC]

Note:
* "new_function_signature" MUST ONLY contain the function name, instead of the full reference to the function. For example, "mean" instead of "numpy.mean".
* Only output the JSON in raw text.

## B.4 Update: Unit Test skeleton

**System prompt:**
You are a very experienced programer. You are good at algorithmic reasoning and writing super high quality code.

The API of interest is:
[OLD_SIGN]
{{{{old_function_signature}}}}
[/OLD_SIGN]

This API recently undergoes an update:
[DESC]

{{{{update_description}}}}
[/DESC]

The API now has the following new function signature:
[NEW_SIGN]
{{{{new_function_signature}}}}
[/NEW_SIGN]

Your task is to write 10 *high-quality* and *comprehensive* unit tests skeletons for testing the validity of the update. A unit test skeleton is a unit test function that only specifies the test inputs. Each unit test skeleton MUST be in raw string, not in Python code block.

Return the set of unit tests skeletons in JSON code block as a list of string. For unit test skeletons generation, following the instructions below:
1: You MUST READ the documentation (between "[DOC]" and "[/DOC]") WORD-BY-WORD and understand it PERFECTLY WELL.
1.1: Also, IDENTIFY important arguments: the more important arguments are ranked to the front in the new function signature.
2: For unit tests, think of a diverse set of API update and the important arguments to test ALL specified behaviors in the documentation — edge-case input, edge-case output, exception raised, etc.
2.1: You need to have different edge-case values for the update and each important arguments (e.g., multi-dimensional input array with different `axis` values).
3: When you generate a new unit test, look CAREFULLY at already generated unit tests, and make sure the inputs are different from previously generated unit tests as much as possible.
3.1: You MUST have proper setup code for API inputs: initialize variables for testing the updated — literally, or randomly generated, etc. INCLUDE in-line comments.
3.2: PREFERABLY, the input to the updated API SHOULD foreseeably lead to a *unique* execution result.
4: The output of the API call MUST be assigned to a variable `result`.
4.1: You MUST call the updated API, instead of old API. If required, you are allowed to call the *old* API by directly calling `old_quad`. ALL other ways to call the old function are FORBIDDEN.
5: If a unit test function is testing throwing exception, you should proceed with `try-except` and finish the unit test function.
5.1: If the test input is meant to testing error catching, check if the API call will raise error. DON'T check error message.
6: If a unit test function is NOT testing throwing exception:
6.1: You MUST output a placeholder `# @ANSWER@` for the right answer to be filled in. Writing the right answer is forbidden.
6.2: Do not write any assertion. This is forbidden. Instead, put a placeholder `# @ASSERT@` at the end of the test function.
6.3: Within the unit function, the placeholders need to start at the left-most indent (i.e. 4 empty spaces — " ").
7: Each test MUST be a function without any input arguments. DON'T attempt to test I/O in each unit tests.
8: The function name MUST be informative. Avoid it to include generic terms like "case1" or "test1".
9: Use "n" as line break. Use 4 empty spaces (" ") as Python code block indent.
10: When you have Python string literal, you MUST use escape for quote — `"` `or `` `; for triple quote — `"""` `or `”` `

**User Prompt:**
This is the documentation that details the behavior about the update:
[DOC]
{{{{update_docstring}}}}
[/DOC]

Only output the set of unit tests skeletons (*a list of strings*) in JSON code block (```json...```).
Include `global {{{{package_name}}}}` as the first line of each unit test function.
If you want to call the old function, you MUST directly call `old_{{{{function_name}}}}`. All other ways to call the old function are FORBIDDEN.

## B.5 Update: Answer generation

**System prompt:**
You are a very experienced programmer. You are good at algorithmic reasoning and writing super high quality code.

The API of interest is
[OLD_SIGN]
{{{{old_function_signature}}}}
[/OLD_SIGN]

This API recently undergoes an update:
[DESC]
{{{{update_description}}}}
[/DESC]

The API now has the following new function signature:
[NEW_SIGN]
{{{{new_function_signature}}}}
[/NEW_SIGN]

You will be given the detailed documentation about the update, and a unit test skeleton with a `# @ANSWER@`. Your task is to generate a Python code block (```python...```) to replace `# @AN-SWER@`. The purpose of the code block is to calculate a value for a variable called `expected_result` or `expected_results`.

For generating the code block, following the instructions below:
1: You MUST READ the documentation (between "[DOC]" and "[/DOC]") WORD-BY-WORD, take a pause and, understand it PERFECTLY WELL.
1.1: Now look at the values of input to the API call, and contemplate on the expected behavior of the *new* API given those inputs.
2: IDENTIFY whether you need to assign value to `expected_result` or `expected_results` — `expected_result` if there's only 1 correct answer; `expected_results` if there's only multiple correct answers. There is only one right choice.
3: Focus on the behavior of the *new* API. When deriving the expected value of `result`, work on this problem STEP-BY-STEP. Then, wisely choose one of the strategies from below:
a. an assignment of a Python literal value to the variable;
b. if the literal is too long or it's best to use arithmetics to get the value, DON'T write literal value. INSTEAD, use step-by-step program code to express how to arrive at the answer.
4: In the code block, DO NOT call the *new* API function. For calculating the answer, you CAN call the *old* API function. However, you MUST directly call `old_quad`. ALL other ways to call the old function are FORBIDDEN.
5: Within the code block, you MUST generate WITH NO leading indent. Use 4 empty spaces (" ") as indent when writing if-else, for-loop, etc.

**User Prompt:**
This is the documentation that details the behavior about the update:
[DOC]
{{{{update_docstring}}}}
[/DOC]

[TEST]
{{{{unit_test_skeleton}}}}
[/TEST]

If you want to call the old function, you MUST directly call `old_{{{{function_name}}}}`. All other ways to call the old function are FORBIDDEN.
{{% if package_instruct %}}
Some special notes for `{{{{package_name}}}}`package:

```
{{{{package_instruct}}}}
{{% endif %}}
```

## B.6 Update: Assertion generation

**System prompt:**
You are a very experienced programer. You are good at algorithmic reasoning and writing super high quality code.

You will be given a unit test function that misses assertion statements to either:
1. check equivalence between `result` and `expected_result`
2. or check equivalence between `result` and any values in `expected_results` ( i.e. multiple correct answer).

Your task is to generate a Python code block (```python...```) to replace `# @ASSERT@`.

**User Prompt:**
[TEST]
{{{{unit_test_skeleton}}}}
[/TEST]

```
{{% if package_instruct %}}
Remember some special features of `{{{{package_name}}}}`package:
{{{{package_instruct}}}}
{{% endif %}}
```

## B.7 Update: Updated Function Implementation

**System prompt:**
You are a very experienced programer. You are good at algorithmic reasoning and writing super high quality code.

The API of interest is
[OLD_SIGN]
{{{{old_function_signature}}}}
[/OLD_SIGN]

This API recently undergoes an update:
[DESC]
{{{{update_description}}}}
[/DESC]

The API now has the following new function signature:
[NEW_SIGN]
{{{{new_function_signature}}}}
[/NEW_SIGN]

And the old API is renamed to:
[OLD_SIGN]
{{{{renamed_old_function_signature}}}}
[/OLD_SIGN]

You will be given the detailed documentation about the update. Your task is to write high quality implementation for the *new* API function in Python code block (```python...```).

To generate the code block, following the instructions below: 1: First of all, you MUST CAREFULLY READ the documentation about the update (between "[DOC]" and "[/DOC]") WORD-BY-WORD and

understand it PERFECTLY WELL.

2: Before arriving at the new implementation, take a deep breath and work on this problem STEP-BY-STEP.

2.1: INCLUDE in-line comments and improve readability. 2.2: If you are provided with unit tests, use them to understand expected behavior of the update.

3: Notice any error handling specified in the documentation. INCLUDE error handling when writing new implementation.

4: The new function's name should be the same as the name in new function signature, with API path removed.

4.1: You MUST NOT write documentation for the new implementation.

4.2: You MUST NOT output the old implementation.

5: To implement the new function, you MUST use the *old* API function AS MUCH AS POSSIBLE.

5.1: Since the bulk part of the functionality is accomplished by the *old* API function, the new implementation MUST be as SUCCINCT as possible.

5.2: You MUST call the *old* API function by directly calling `old_quad`. ALL other ways to call the old function are FORBIDDEN.

6: DO NOT write imports.

7: Use 4 empty spaces (" ") as Python code block indent.

**User Prompt:**

This is the documentation that details the behavior about the update:

[DOC]

{{{{update_docstring}}}}

[/DOC]

{{% if unit_tests %}}

Unit tests for new update:

[PYTHON]

{{%- for test in unit_tests %}}

# Unit Test {{{{loop.index}}}}

{{{{test}}}}

{{% endfor -%}}

[/PYTHON]

{{% endif %}}

If you want to call the old function, you MUST directly call `old_{{{{function_name}}}}`. All other ways to call the old function are FORBIDDEN.

You MUST NOT output the old implementation.

You MUST NOT implement `old_{{{{function_name}}}}`.

Only output the new implementation in Python code block (```python...```).

---

## B.8 Generate missing import

**System prompt:**

You are a very experienced programer. You are good at algorithmic reasoning and writing super high quality code.

Your task is to write import statements to include any package dependency before running the code. Return import statements in Python code block (```python...```).

To generate the code block, following the instructions below:

1: First of all, read the code WORD-BY-WORD and understand it PERFECTLY WELL.

2: DO NOT miss type hints in function signature, function body, etc.

3: If no import statements is required, output an empty Python code block.

**User Prompt:**

[PYTHON]

{{code}}

[/PYTHON]

Only output the Python code block (```python...```).

## B.9 Package Instruction

**`re`: Assertion generation:**
1: To compare `re.Match` object, `==` doesn't work. One should use `group()` method to obtain the string and then compare, e.g. `m1.group() == m2.group()`.
2: When no match is found, the output will be None. Make sure this situation is dealt with.

**`torch`: Assertion generation:**
1: Using `==` to check equality of Tensor objects (e.g. numpy.array) is ambiguous. For example, you should use `torch.equal` or `torch.allclose` to check if two Tensor objects equal.
1.1: allclose(): argument 'input' (position 1) must be Tensor, not list.

**`itertools`: Assertion generation:**
1: The output of `itertools` functions (e.g. `itertools._grouper` object) is not directly checkable by `==`. To compare the output of itertools, the most direct way is to unwrap the output into something directly checkable (e.g. list, tuple, dict).

**`itertools`: Answer generation. For itertools.groupby only:**
If you make call to `old_groupby`, don't attempt to unwrap the function output (e.g. by list()).

**`numpy`: Assertion generation:**
1: Using `==` to check equality of numpy objects (e.g. numpy.array) is ambiguous. For example, you should use `numpy.equal` or `numpy.allclose` to check if two numpy array equal.

## B.6 GENERATION PROMPT: PROGRAM SYNTHESIS

See Prompt B.10 for generating program synthesis specifications

See Prompt B.11 for generating unit test skeletons

See Prompt B.12 for generating unit test answers; part of the prompt takes corresponding instruction from Prompt B.9 to guide the model to generate for different packages.

See Prompt B.13 for generating unit test assertions; part of the prompt takes corresponding instruction from Prompt B.9 to guide the model to generate for different packages.

See Prompt B.14 for generating reference solutions that use the updated function.

## B.10 ProgSyn: Problem specification

**System prompt:**
You are a helpful assistant. You think deeply and creatively. Your task is to think of and write interesting tutorial(s) for an API update. mainly <problem, solution>.

You will be given the full information about an update to an existing Python package. You should think of usage (i.e. program synthesis example) of the updated API signature that satisfy the following criteria:
* the problem scenario posed by the program synthesis example MUST follow the general functionality of the (old and new) API.
* the problem scenario MUST be affected and preferably benefited by the API update. By benefit, it means the code complexity of the solution will be reduced.
* the problem MUST be at least medium hard, so that the solution MUST make *non-trivial* use of the API's functionality.
* Be given the number of parameters that the solution accepts.

Return the entire response in JSON format as a dictionary. Make sure nested brackets are closed correctly. Be careful with unterminated string literal. The dictionary should contain the following:
1: "scenario": (as string) a real-world scenario that the problem is situated in. Keep it medium short.
1.1: Avoid including information – e.g. exact term – about API changes, or package needs to be used in "problem".

2: "problem": (as string) problem specification that needs solving by a Python function. Keep it short.
2.1: Avoid giving imperative instruction on how to solve the problem. MUST Remain at high-level. Avoid including information – e.g. exact term – about API changes, or package needs to be used in "problem".
2.2: Make sure the description of the input is well connected and blend into the description of the scenario.
2.3: Design the problem such that each input to the solution is meaningfully used in the code.
3: "solution_signature": (as string) the function signature of the solution function.
3.1: the function name should be derived from "scenario".

Give me 1 diverse program synthesis example(s).

**User Prompt:**
In Python package `{{package_name}}`, there's an API function `{{api_path}}` as follows: [OLD_SIGN] {{old_func}} [/OLD_SIGN]

Maintainer of the package thinks it's best to introduce the following update
[DESC]
{{update_description}}
[/DESC]

This is because
[RATIONALE]
{{update_rationale}}
[/RATIONALE]

The function docstring now differs with previous version in the following way:
[DOC]
{{docstring_diff}}
[/DOC]

And the function has the following new signature:
[NEW_SIGN]
{{new_function_signature}}
[/NEW_SIGN]

The problem *MUST* non-trivially benefit from the update (i.e. new API); so that solving the problem with the old API is not possible, or requires more efforts (e.g. need to write longer code). The solution of the problem must accept {{num_param}} parameter(s).

Note:
Only output the JSON in raw text.

---

### B.11 ProgSyn: Unit Test Skeleton

**System prompt:**
You are a very experienced programer. You are good at algorithmic reasoning and writing super high quality code.

Your task is to write 10 *high-quality* and *comprehensive* unit tests skeletons for testing validity of any solution function to a problem specification. A unit test skeleton is a unit test function except the right answer being clearly specified. Each unit test skeleton MUST be in raw string, not in Python code block.

Return the set of unit tests skeletons in JSON code block as a list of string. For unit test skeletons generation, following the instructions below:
1: You MUST READ the problem specification (between "[PROBLEM]" and "[/PROBLEM]") WORD-BY-WORD and understand it PERFECTLY WELL.
1.1: Also, IDENTIFY important arguments: the more important arguments are ranked to the front in the new function signature.

2: For unit tests, READ the scenario description (between [SCENARIO]...[/SCENARIO]) WORD-BY-WORD and understand it PERFECTLY WELL.

2.1: Contemplate, and think of a diverse set of representative inputs to solution function; this set of input should capture possible and interesting cases which solution function might encounter after deployment.

2.2: BE SURE to test ALL specified behaviors in the problem specification — edge-case input, edge-case output, exception raised, etc.

2.3: You need to have different edge-case values for the update and each important arguments (e.g., multi-dimensional input array with different `axis` values).

3: When you generate a new unit test, look CAREFULLY at already generated unit tests, and make sure the inputs are different from previously generated unit tests as much as possible.

3.1: You MUST have proper setup code for solution function inputs: initialize variables for testing the updated — literally, or randomly generated, etc. INCLUDE in-line comments.

3.2: PREFERABLY, the input to the solution function call SHOULD foreseeably lead to a *unique* execution result.

4: The output of the solution function MUST be assigned to a variable `result`.

4.1: You MUST call the solution function.

5: If a unit test function is testing throwing exception, you should proceed with `try-except` and finish the unit test function.

5.1: If the test input is meant to testing error catching, check if the API call will raise error. DON'T check error message.

6: If a unit test function is NOT testing throwing exception:

6.1: You MUST output a placeholder `# @ANSWER@` for the right answer to be filled in. Writing the right answer is forbidden.

6.2: Do not write any assertion. This is forbidden. Instead, put a placeholder `# @ASSERT@` at the end of the test function.

6.3: Within the unit function, the placeholders need to start at the left-most indent (i.e. 4 empty spaces — " ").

7: Each test MUST be a function without any input arguments. DON'T attempt to test I/O in each unit tests.

8: The function name MUST be informative. Avoid it to include generic terms like "case1" or "test1".

9: Use "
n" as line break. Use 4 empty spaces (" ") as Python code block indent.

**User Prompt:**

In a real-world scenario, there exists some trouble to be solved:

[SCENARIO]

{{{{scenario}}}}

[/SCENARIO]

Luckily, someone could solve this trouble by writing a function, as long as the solution function satisfy the following problem specification:

[PROBLEM]

{{{{problem}}}}

[/PROBLEM]

Additionally, the solution function should have the following function signature:

[SOLUTION_SIGN]

{{{{solution_signature}}}}

[/SOLUTION_SIGN]

{{% if package_instruct %}}

Some special notes for `{{{{package_name}}}}` package:

{{{{package_instruct}}}}

{{% endif %}}

Only output the set of unit tests skeletons (*a list of strings*) in JSON code block (```json...```).

## B.12 ProgSyn: Answer generation

**System prompt:**
You are a very experienced programer. You are good at algorithmic reasoning and writing super high quality code.

In a real-world scenario, there exists some trouble to be solved:
[SCENARIO]
{{{{scenario}}}}
[/SCENARIO]

Luckily, someone could solve this trouble by writing a function, as long as the solution function satisfy the following problem specification:
[PROBLEM]
{{{{problem}}}}
[/PROBLEM]

An ideal solution function takes the following function signature:
[SOLUTION_SIGN]
{{{{solution_signature}}}}
[/SOLUTION_SIGN]

You will be a unit test skeleton with a `# @ANSWER@`. Your task is to generate a Python code block (```python...```) to replace "`# @ANSWER@". The purpose of the code block is to calculate a value for a variable called `expected_result` or `expected_results`.

For generating the code block, following the instructions below:
1: You MUST READ the problem specification (between "[PROBLEM]" and "[/PROBLEM]") WORD-BY-WORD, take a pause and, understand it PERFECTLY WELL.
1.1: Now look at the values of input to the solution function, and contemplate on the expected behavior of the solution function given those inputs.
2: IDENTIFY whether you need to assign value to `expected_result` or `expected_results`. There is only one right choice.
3: Before arriving at an answer, ALWAYS take a deep breath and work on this problem STEP-BY-STEP. Then, wisely choose one of the strategies from below:
a. an assignment of a Python literal value to the variable;
b. if the literal is too long or it's best to use arithmetics to get the value, DON'T write literal value. INSTEAD, use step-by-step program code to express how to arrive at the answer.
4: Within the code block, you MUST generate WITH NO leading indent. Use 4 empty spaces (" ") as indent when writing if-else, for-loop, etc.

**User Prompt:**
To write code to calculate `expected_result` or `expected_results` (strategy b), maybe the following two functions are useful:

The first function comes from package `numpy`.
[FUNCTION1]
{{{{old_function_signature}}}}
[/FUNCTION1]

The second function is an updated version of the FUNCTION1
[FUNCTION2]
{{{{new_function_signature}}}}
[/FUNCTION2]

FUNCTION2 differs from FUNCTION1 in the following way:
[DOC]
{{{{update_docstring}}}}
[/DOC]

[TEST]
{{{{unit_test_skeleton}}}}
[/TEST]
{{% if package_instruct %}}

Some special notes for `{{{{package_name}}}}`package:
{{{{package_instruct}}}}
{{% endif %}}

## B.13 ProgSyn: Assertion generation

**System prompt:**
You are a very experienced programer. You are good at algorithmic reasoning and writing super high quality code.

You will be given a unit test function that misses assertion statements to either:
1. check equivalence between `result`and `expected_result`
2. or check equivalence between `result`and any values in `expected_results`( i.e. multiple correct answer).

Your task is to generate a Python code block (```python...```) to replace `# @ASSERT@`.

**User Prompt:**
[ TEST]
{{{{unit_test_skeleton}}}}
[ /TEST]

{{% if package_instruct %}}
Remember some special features of `{{{{package_name}}}}`package:
{{{{package_instruct}}}}
{{% endif %}}

## B.14 ProgSyn: Solution

**System prompt:**
You are a very experienced programer. You are good at algorithmic reasoning and writing super high quality code.

The API of interest is
[OLD_SIGN]
{{{{old_function_signature}}}}
[/OLD_SIGN]

This API recently undergoes an update and it now has the following new function signature:
[NEW_SIGN]
{{{{new_function_signature}}}}
[/NEW_SIGN]

This is the documentation that details the behavior about the update:
[DOC]
{{{{update_docstring}}}}
[/DOC]

You will be given the detailed problem specification. Your task is to USE the new API (between "[NEW_SIGN]" and "[/NEW_SIGN]") to write high quality solution function that solve the problem specification in Python code block (```python...```).

To generate the code block, following the instructions below:
1: First of all, you MUST CAREFULLY READ the problem specification (between "[PROBLEM]" and "[/PROBLEM]") WORD-BY-WORD and understand it PERFECTLY WELL.
2: Before arriving at the solution function, take a deep breath and work on this problem STEP-BY-STEP.
2.1: INCLUDE in-line comments and improve readability.
2.2: If you are provided with unit tests, use them to understand expected behavior of the solution function.
3: Notice any error handling specified in the problem specification. INCLUDE error handling when writing solution.
4: The solution signature MUST follows the one specified between "[SOLUTION_SIGN]" and "[/SOLUTION_SIGN]".

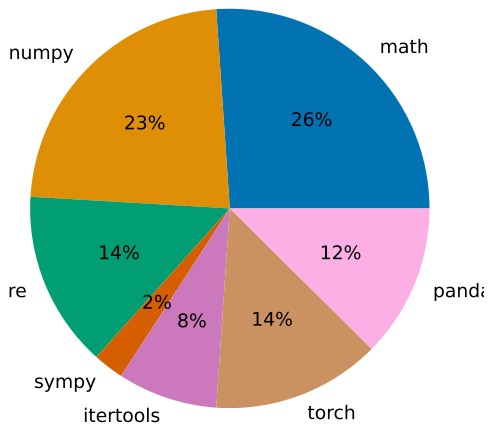

Figure 5: Package breakdown of updated functions in `CodeUpdateArena`

> 4.1: You MUST NOT write documentation for the solution.
> 5: To implement the solution, you MUST use the *new* API function AS MUCH AS POSSIBLE.
> 6: Use 4 empty spaces (" ") as Python code block indent.
>
> **User Prompt:**
> [PROBLEM]
> {{{{problem}}}}
> [/PROBLEM]
>
>
> Solution should take the following singautre
> [SOLUTION_SIGN]
> {{{{solution_signature}}}}
> [/SOLUTION_SIGN]
> {{% if unit_tests %}}
> Unit tests for new update:
> [PYTHON]
> {{%- for test in unit_tests %}}
> # Unit Test {{{{loop.index}}}}
> {{{{test}}}}
> {{% endfor -%}}
> [/PYTHON]
> {{% endif %}}
> USE the new API (between "[NEW_SIGN]" and "[/NEW_SIGN]") to write high quality solution function that solve the problem specification in Python code block (```python...```). Only output the new implementation in Python code block (```python...```).

## C ADDITIONAL DATASET STATISTICS

### C.1 RAW STATISTICS

Figure 5 shows the fraction of examples in our dataset per package.

Figure 6 shows the number of examples per update type in our dataset.

Figure 7 shows the number of program synthesis examples per API update. All updates have at least 3 examples, with some having substantially more if diverse enough samples could be drawn.

Table 8 shows that our benchmark covers a range of different types of API functionalities.

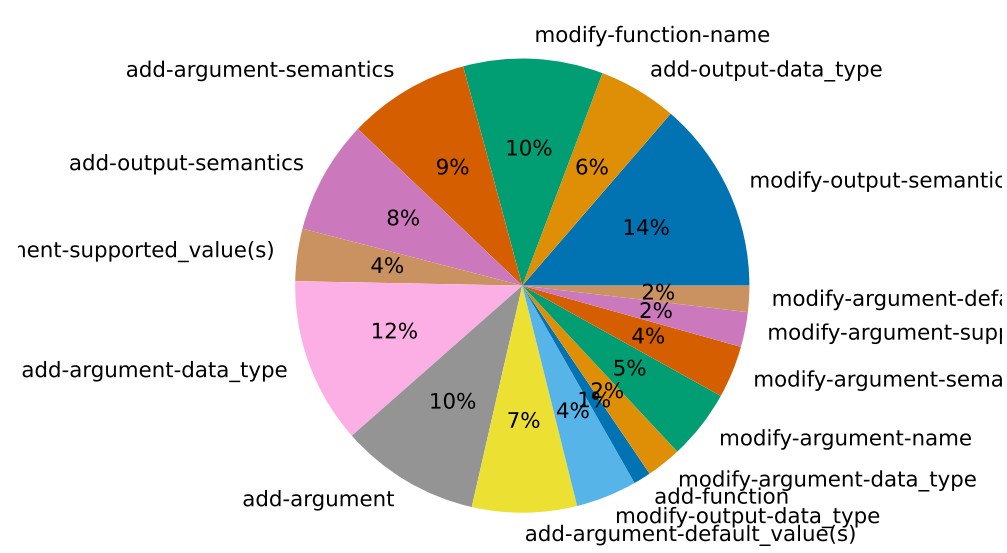

Figure 6: Distribution of Program Synthesis examples covered by different update types.

### C.2 HUMAN INSPECTION OF FAILED GPT-4 ATTEMPTS

Taking the predicted solution in Table 3, we manually inspect 330 predicted solutions from 66 PS examples where GPT-4 failed to generate *a correct solution*.[5] We categorize errors into 4 categories. (1) Incomplete Solution: includes issues like failure to include the edge cases of the problem statement, missing or incorrect library imports, and incorrectly thrown exceptions as per the problem statement. (2) Wrong Solution: real mistakes due to misinterpretation of the problem statement, using incorrect semantics for mathematical computation, etc. (3) Wrong Test Case: test cases are incorrect or cover cases not expected from the problem statement. (4) Specification Error: the specification was not complete enough for the model to choose the right output. Table 9 shows the breakdown across these categories. We note that an example may have error in multiple places (e.g. unit tests, predicted solution, etc.), and therefore the error categories are not mutually exclusive.

Table 8: Diversity of packages in `CodeUpdateArena`. Our benchmark covers a range of different types of API functionalities. Our python version is `3.11.5`.

| Package | Type | Standard / External lib. |
|---|---|---|
| `re` | String operations | Standard |
| `math` | Arithmetic operations | Standard |
| `itertools` | Python data structure operations | Standard |
| `torch==2.0.1, numpy==1.25.2` | Vector operations | External |
| `sympy==1.12` | Symbolic operations | External |
| `pandas==2.1.0` | Table operations | External |

---

[5]The inspection was conducted on a preliminary version of the benchmark not including pandas.

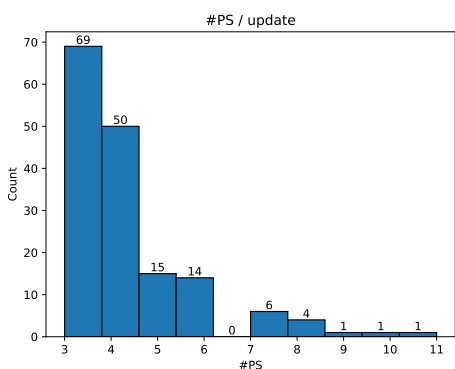

Figure 7: Number of program synthesis instances per API update in `CodeUpdateArena`.

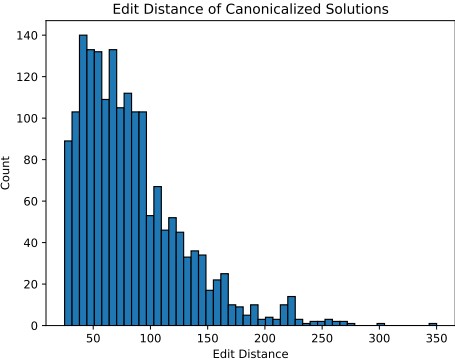

Figure 8: Edit distances between canonicalized reference solutions of PS instances; pairing happens among PS of a single update.

### C.3 TEST COVERAGE

We conducted a line coverage analysis with package `coverage` by running all the unit tests on the reference solution. We heuristically exclude lines and find that our test coverage is high: if we exclude function definition (i.e. "def") and imports (i.e. "import"), our line coverage is $83.6\%$. Since we do not test for specific errors being thrown, excluding lines containing "except" and "raise" results in a coverage rate of $97.0\%$.

Table 9: Manual categorization of 66 failures cases of GPT-4 on program synthesis examples. Categories are not exclusive.

| Error Category | Count |
| --- | --- |
| Incomplete Solution | 29 |
| Wrong Solution | 33 |
| Wrong Test Case | 13 |
| Specification Error | 2 |

## D IMPLEMENTATION DETAILS OF COMPUTING `UPass@k`

As described in Section 3, to evaluate each predicted solution $\tilde{c}_i$, our evaluation procedure executes the set of test cases *twice*, once with the updated API and once with the old API.

```
Dist 15

Prog A:
CANONICALIZED
import math
from typing import Tuple

def var0(var1, var2):
    return math.nextafter(var1, var2)
Prog B:
CANONICALIZED
import math
from typing import Tuple

def var0(var1, var2):
    var3, var4 = math.nextafter(var1, var2)
    return (var3, var4)
```

```
Dist 23
Prog A:
CANONICALIZED
from typing import List, Union
import math

def var0(var1, var2):
    var3 = []
    for var4 in var1:
        var5 = math.sqrt(var4, fallback=var2)
        var3.append(var5)
    return var3
Prog B:
CANONICALIZED
from typing import List
import math

def var0(var1, var2):
    var3 = [math.sqrt(var4, fallback=var2) for var4 in var1]
    return var3
```

Figure 9: Example of reference solution with low edit distance

```
# [imports]
import numpy
...
# [implementation of updated API]
def argsort(..., reverse=False):
    ...
# [Optional: update API at runtime]
setattr(numpy, "argsort", argsort)
# [Predicted solution]
...
# [Unit test function]
def test_reverse_false():
    ...
test_reverse_false()
```

Figure 10: Example of test execution

Table 10: Hyperparameter search over number of gradient updates when FT(U) continues pretraining on the update docstring. We found that our choice of 10 in Table 4 is optimal. In this experiment, other hyperparameters are kept the same, including the constant learning rate schedule and learning rate of 1e-3.

| DS-Coder-v1.5 | | UPass (Efficacy) ↑ | | SPass (Specificity) ↑ | |
| Method | #gradient update | @1 ($\Delta$) | @5 ($\Delta$) | @1 ($\Delta$) | @5 ($\Delta$) |
|---|---|---|---|---|---|
| | 2 | 3.4 | 7.0 | 49.5 | 71.3 |
| FT (U) | 5 | 3.4 | 7.0 | 49.4 | 68.7 |
| | 10 | 3.6 | 7.0 | 56.4 | 77.3 |

To evaluate code conforming to a new, non-standard API, we use a setup as shown in Figure 10. We put the implementation of the new API (e.g. `argsort`) at the top of the program (after `imports`). Then, we follow by a simple statement of `setattr(numpy, "argsort", argsort)` to dynamically rebind the reference of `numpy.argsort` (old API) to the new API.

Given the total number of trials $n$, the target value $k$, and the number of successes $c_i$ on example $i$ (pass tests and use the update), we compute UPass@k over $D$ program synthesis examples using the same form as in Chen et al. (2021):

$$\text{UPass@k} = \frac{1}{D} \sum_{i=1}^{D} \left[ 1 - \frac{\binom{n-c}{k}}{\binom{n}{k}} \right].$$

Finally, note that when performing our editing updates, each example is updated independently; a update $u'$ starts again from the base model $\mathcal{M}$.

# E    EXPERIMENTATION SETUP

## E.1    HYPERPARAMETERS

In Table 10, for FT(U), we conducted a hyper-parameter search over the number of gradient update when training on the documentation about the API update. More training does not necessarily lead to degradation in specificity.

```
# training hyper
# if hypers are unspecified, the values are set to be the default in `transformers`
optimizer: adamw_torch # as defined in TrainingArgument in `transformers`
lr: 1e-3
lr_scheduler_type: constant # in preliminary study, we found using `linear` leads to
↪   worse performances
batch_size: min(train_set_size, 8)
num_epoch: # 10 for FT(U) , and 5 for FT(PS)
decay: 1e-8
warmup_ratio: 0.05
gradient_accumulation_steps: 1
```

```
# Generation:
do_sample: True
top_p: 0.7
temperature: 0.8
# Control the length of generation
max_new_tokens: 512
```

## E.2    LoRA CONFIGURATION

```
# lora
r: 8 # the low rank dim (hidden->r->hidden )
alpha: 1
dropout: 0.1
```

Table 11: Experiments on DS-Coder-v1 with different training set construct controlled by $(c, r)$. Despite to a lesser degree, the observations in Table 5 hold for DS-Coder-v1 as well — training on unrelated examples is worse, but including random updates along with the true updates help. *: comparing against base model, the gap is significant according to a paired bootstrap test with $p < 0.05$.

| | | | UPass (Efficacy) ↑ | | SPass (Specificity) ↑ | |
|---|---|---|---|---|---|---|
| Method | $c$ | $r$ | @1 ($\Delta$) | @5 ($\Delta$) | @1 ($\Delta$) | @5 ($\Delta$) |
| DS-Coder-v1 | — | — | 2.6 | 4.3 | 49.3 | 79.3 |
| + Prepend | — | — | $10.7^*_{+8.1}$ | $18.6^*_{+14.3}$ | — | — |
| + FT (PS) | 2 | 1 | $30.8^*_{+28.2}$ | $49.7^*_{+45.3}$ | $52.5^*_{+3.3}$ | $78.4_{-0.8}$ |
| | 1 | 2 | $25.7^*_{+23.1}$ | $45.3^*_{+41.0}$ | $53.3^*_{+4.0}$ | $79.8_{+0.5}$ |
| | 3 | 0 | $30.4^*_{+27.8}$ | $41.6^*_{+37.3}$ | $51.6^*_{+2.3}$ | $77.0^*_{-2.2}$ |
| | 0 | 3 | $8.3^*_{+5.7}$ | $18.0^*_{+13.7}$ | $51.5^*_{+2.2}$ | $79.1_{-0.1}$ |

```
# where the lora are inserted:
target_modules = ["q_proj", "v_proj"]
```

### E.3 Evaluation procedure for finetuning experiments

Recall that our benchmark `CodeUpdateArena` is structured by pairing each executable API update with $n$ program synthesis examples. We treat the $n$ program synthesis examples as an ordered list. The training sets of FT (U) and FT (PS) slightly differ and we will describe them separately.

**FT (U)** The training set only consists of a single copy of the API update information following the template in Prompt E.4.

**FT (PS)** As mentioned in Section 6.1, the single-edit training set contains $c$ copies of $N_u = 2$ unique program synthesis examples and $N_r$ examples from $r$ updates from the rest of the benchmark. Since the number of program synthesis examples per update could be as few as 3, we adopt a cross-validation scheme to evaluate a model for each update. To give a concrete example: when testing the model on program synthesis example $i \in [0, n)$, we take the "previous" $N_u$ examples — example $(i - 1) \mod n$ and example $(i - 2) \mod n$; we then repeat them $c$ times to obtain the final set of examples for target update. Then, we take *two* unique program synthesis examples from each of the $r$ random updates; and combine them with examples from the previous step to obtain the final training set. Each training instance is formatted with Prompt E.5.

### E.4 Additional ablation study

See Figure 11 for a specific study for $c$.
See Figure 12 for a specific study for $r$.

### E.5 Compute Resources

For GPT-4, we call through the `openai` Python interface. It takes about 2 hours to generate (5) solutions to program synthesis examples. For open-source models, all our experiments are accomplished on NVIDIA A40 with 48GB memory. In our work, each experiment (prepend and fine-tuning) takes a max of 9.5 hours to finish generating (5) solutions to program synthesis examples. After generating predicted solutions to program synthesis examples, we need to execute the generated program against corresponding test cases. This process is CPU-only and finishes within 2 hours.

### E.6 Prompt

Our prompt mostly migrates the style of the ones used in CodeLlama Rozière et al. (2023).

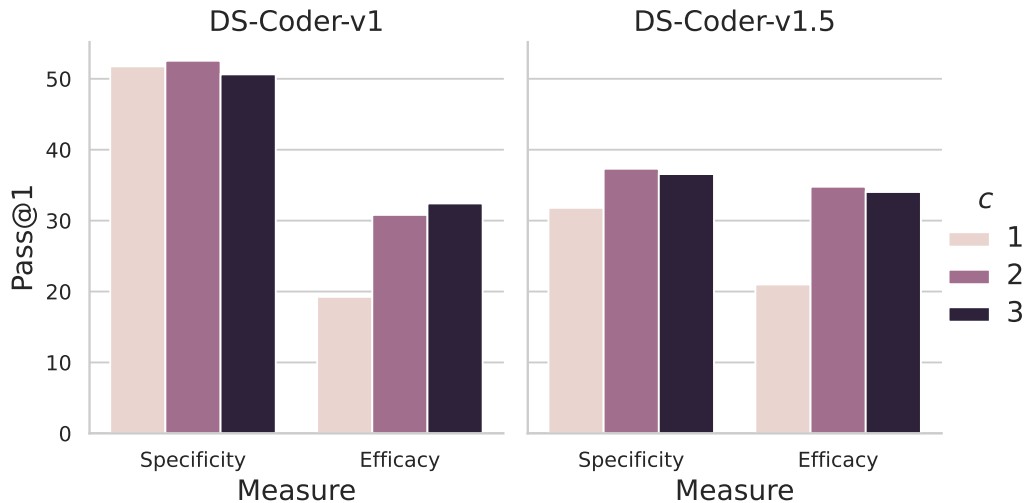

Figure 11: Ablation study of $c$ — the number of times to repeat $N_u$ unique program synthesis examples from target update. $r$ is fixed to be 1. We observed that $c = 2$ is the optimal hyper, beyond which we observe diminishing gains in efficacy and drops in specificity.

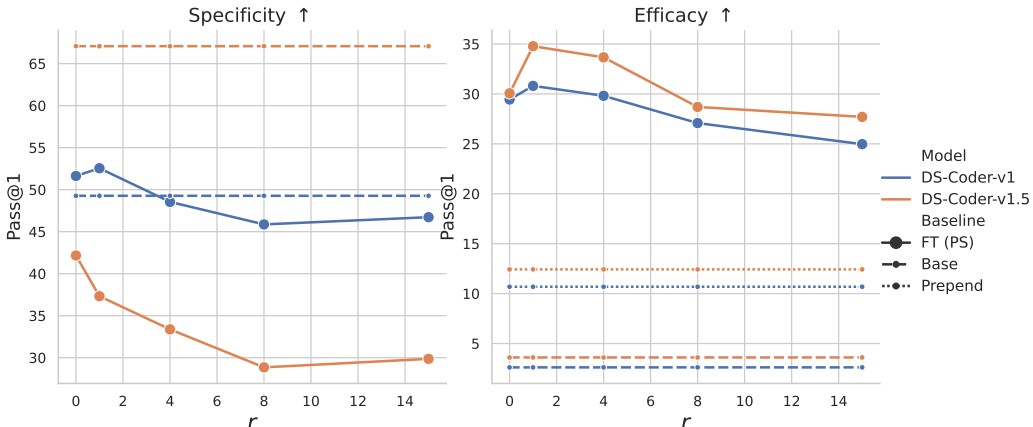

Figure 12: Ablation study of $r$ — the number of random updates where a pair of unique program synthesis examples are drawn from each update. $c$ is fixed to be 2. Although different models have different optimal values, we found that larger $r$ will continue to decrease models' performances for efficacy and specificity.

See Prompt E.1 for HumanEval

See Prompt E.2 for template for base model experiment

See Prompt E.3 for template for prepending experiment

See Prompt E.4 for template to generate instance for FT(U)

See Prompt E.5 for FT(PS)

### E.1 HumanEval in jinja2

```
[INST]
Please continue to complete the function. You are not allowed to modify the given code and do the completion
only. Please return all completed function in [PYTHON] and [/PYTHON] tags. Here is the given code to do
completion:
[PYTHON]
{{completion_context}}
[/PYTHON]
[/INST]
```

### E.2 Base Model

```
[INST]
Your task is to write a Python solution to a problem in a real-world scenario.
The Python code must be between [PYTHON] and [/PYTHON] tags.

Scenario: {{example_scenario}}
Problem: {{example_problem}}
Solution signature: {{example_solution_signature}}
[TEST]
{{example_unit_tests}}
[/TEST]
[/INST]

[PYTHON]
{{example_solution}}
[/PYTHON]

[INST]
Scenario: {{scenario}}
Problem: {{problem}}
Solution signature: {{solution_signature}}
[TEST]
{{unit_tests}}
[/TEST]
[/INST]
```

### E.3 Prepend in jinja2

```
[INST]
Update note:
There's an recent update to a function `{{old_function_signature}}`— {{update_description}}. The
function now has a new function signature — `{{new_function_signature}}`.
Here's a detailed documentation about the update:
[DOC]
{{update_docstring}}
[/DOC]

Your task is to write a Python solution to a problem in a real-world scenario.
The Python code must be between [PYTHON] and [/PYTHON] tags.

Scenario: {{example_scenario}}
Problem: {{example_problem}}
```

```
Solution signature: {{example_solution_signature}}
[TEST]
{{example_unit_tests}}
[/TEST]
[/INST]

[PYTHON]
{{example_solution}}
[/PYTHON]

[INST]
Scenario: {{scenario}}
Problem: {{problem}}
Solution signature: {{solution_signature}}
[TEST]
{{unit_tests}}
[/TEST]
[/INST]
```

## E.4 FT(U) in jinja2

**Train:**
```
[INST]
Update note:
There's an recent update to a function `{{old_function_signature}}`— {{update_description}}. The
function now has a new function signature — `{{new_function_signature}}`.

Here's a detailed documentation about the update:
[DOC]
{{update_docstring}}
[/DOC]
[/INST]
```

**Evaluation:**
```
[INST]

{% if include_update -%}
Update note:
There's an recent update to a function `{{old_function_signature}}`— {{update_description}}. The
function now has a new function signature — `{{new_function_signature}}`.
Here's a detailed documentation about the update:
[DOC]
{{update_docstring}}
[/DOC]
{% endif %}

Your task is to write a Python solution to a problem in a real-world scenario.
The Python code must be between [PYTHON] and [/PYTHON] tags.

Scenario: {{example_scenario}}
Problem: {{example_problem}}
Solution signature: {{example_solution_signature}}
[TEST]
{{example_unit_tests}}
[/TEST]
[/INST]
[PYTHON]
{{example_solution}}
[/PYTHON]

[INST]
```

```
Scenario: {{scenario}}
Problem: {{problem}}
Solution signature: {{solution_signature}}
[TEST]
{{unit_tests}}
[/TEST]
[/INST]
```

### E.5 FT(PS) in jinja2

```
{: Train and Evaluation} [INST]
{% if include_update -%}
Update note:
There's an recent update to a function `{{old_function_signature}}`— {{update_description}}.  The
function now has a new function signature — `{{new_function_signature}}`.
Here's a detailed documentation about the update:
[DOC]
{{update_docstring}}
[/DOC]
{% endif %}

Your task is to write a Python solution to a problem in a real-world scenario.
The Python code must be between [PYTHON] and [/PYTHON] tags.

Scenario: {{example_scenario}}
Problem: {{example_problem}}
Solution signature: {{example_solution_signature}}
[TEST]
{{example_unit_tests}}
[/TEST]
[/INST]

[PYTHON]
{{example_solution}}
[/PYTHON]

[INST] Scenario: {{scenario}}
Problem: {{problem}}
Solution signature: {{solution_signature}}
[TEST]
{{unit_tests}}
[/TEST]
[/INST]
```

## F  LICENSING

We use the following open-source LLMs with open licenses.

**CODELLAMA**   Rozière et al. (2023) uses the LLAMA 2 COMMUNITY LICENSE (see `https://github.com/meta-llama/codellama/`).

**DEEPSEEKCODER**   DeepSeek-AI et al. (2024) uses DEEPSEEK LICENSE (see `https://github.com/deepseek-ai/DeepSeek-Coder/`).

**DEEPSEEKCODER-V1.5**   Guo et al. (2024a) uses the DEEPSEEK LICENSE (see `https://github.com/deepseek-ai/DeepSeek-Coder/`).

