# OpenReview forum: "CodeUpdateArena: Benchmarking Knowledge Editing on API Updates"
_ICLR.cc/2025/Conference — Submitted to ICLR 2025_

### Official Review · Reviewer_jDUt · 2024-11-03

**Soundness:** 3
**Presentation:** 3
**Contribution:** 3
**Rating:** 8
**Confidence:** 4

**Summary:**

This paper presents a new program synthesis dataset focused on evaluating how well an LLM can be adapted to handle changes to an existing API. In each task, an LLM gets an API change description and a few example usages, and after after fine-tuning (or otherwise integrating this information) must solve synthesis examples using the new API, without being prompted with the new API. The evaluation metric ensures that the new API is used and that the old API wouldn't have worked in place of it. The benchmark itself is LLM generated, but with considerable testing and manual inspection for quality control. The evaluation finds that while fine tuning on updated API examples improves performance a bit, none of the evaluated fine tuned models achieves high accuracy. They also ensure the solvability of the dataset by showing that in a few-shot setting with the API functions provided, most of the benchmark is solved.

**Strengths:**

The dataset created in this work deals with an interesting and real-world applicable problem, one that is particularly relevant to large language models which are trained on large amounts of historical code. Broadly I think that such a dataset is a good thing for the community to have, and that it will encourage future research in this area – and this work does a good job of building and evaluating the dataset.

The metrics presented in the work, UPass@K (performance on problems using new API) and SPass@K measuring specificity (i.e. performance degradation on unrelated problems), are solid metrics for evaluating the task. In particular it's important that UPass@K ensures that the new solution not only passes all the unit tests but does so using the new functionality, and that swapping in the old functionality would *not* solve all the tests. Overall this work does a good job of setting up a sensible and precise problem statement / set of metrics for future work to be evaluated on.

While an LLM is used for dataset generation, considerable effort was put into automatically and manually screening the generated tasks – both for the API specifications (4.2) and for ensuring that solving failures in the few-shot setting are not because of incorrect specifications (Table 3 and Appendix C.2).

The evaluation is reasonable for a paper presenting this dataset – a novel approach to solving the problem isn't presented, but doesn't need to be for a dataset paper in a new, compelling domain.

**Weaknesses:**

- I'm confused about how the Base Models could perform so well, which I've elaborated on in the Question section – this is my main concern since it suggests there's something I'm misunderstanding either about those Base Models or the dataset itself.
- (minor) Including the c=1 (so r=2) condition in ablation Table 5 would be nice to see, just to be sure that c=2 really provides the best performance benefit before it starts to drop with c=3.
- Quick fix – Table 8's description should say 77 failures not 66, at least that's what the column adds up to.

> We observed relatively few cases of incorrect test cases or bad specifications, indicating that our dataset is of sufficient quality to test knowledge editing methods.

- From the appendix it seems to be that 19.5% (15/77) of failures are for these reasons, which is not ideal but okay – manual inspection to fix these would be nice for verifying the quality of the dataset. But not a big problem given what a small fraction this is of the total dataset (since 19.5% is only within failures, which themselves are 16.9% of all the problems so it's only about 3% of the whole dataset, if I understand correctly).

**Questions:**

- I would have expected the Base Model row of Table 4 to get essentially 0% for UPass for all models, since UPass requires not only solving the problem but actually solving it using new API in a way that would not work with the old API – but instead models get as high as 22.7% in Pass@1 and 33.3% in Pass@5. My understanding is that the Base Model doesn't get to see the new API at all (neither in fine tuning nor prompting), so in order to solve any problems it would have to somehow hallucinate the correct API change, misunderstanding the old API to behave like the new API despite having never heard about the new API. It seems really surprising that that would happen so often.
  - Am I understanding the Base Model correctly? If so, can you explain this, perhaps providing examples of where this happens to help understand how it could be so common? How could models solve anything without having heard of the new API function?

- In section 6 it's mentioned that 2 program synthesis problems are used for fine tuning, because many update functions only have 3 synthesis problems so the remaining one is held out for testing (and cross validation is used). In cases where where there are more than 3 synthesis problems, what does it mean to be "correct" on this update specification. Does the model need to solve *all* of the held out synthesis problems in one go (with k=1 or k=5 attempts)? Or does it only need to solve one at a time, and these are treated as separate entries in the dataset (where each entry has 1 test synthesis problem and 2 training synthesis problems)?

---

> ### Author Response · Authors · 2024-11-26
> **Rebuttal**
>
> **[Weakness 1 / Question 1]  High performance of Base model**
>
> See general response.
>
> **[Question 2] Question about evaluation procedure.**
>
> Your second interpretation is correct — “it only needs to solve one at a time, and these are treated as separate entries in the dataset (where each entry has 1 test synthesis problem and 2 training synthesis problems)”
>
> **[Weakness 2] Missing a minor baseline ablation of c=1 and r=2.**
>
> We ran the experiment and we found that the default setting of (c, r) = (2, 1) in our paper remains the optimal choice. See Table 5 and 11 of the updated draft.
>
> **[Weakness 3] The number of annotated failure cases instances in Table 8 (in previous draft).**
>
> We thank the reviewer for the very careful read. We examined 66 failed program synthesis examples, but each could have been assigned more than one failure reason. We added the clarification to Line 1764.
>
> **[Weakness 4] Scale of erroneous examples in the dataset**
>
> Yes, it’s only about 3% of the whole dataset at time of inspection.

---

> ### Author Response · Authors · 2024-12-03
> **Reminder about end of discussion period**
>
> Dear Reviewer,
>
> As we approach the response deadline, we wanted to kindly check if our revisions have adequately addressed your concerns. If you have any remaining questions or concerns, we would be happy to address them.
>
> Thank you for your time and consideration.
>
> Best regards,
>
> Authors

---

> > ### Comment · Reviewer_jDUt · 2024-12-03
> >
> > Thank you for the responses and for fixing the bug – that resolves my confusion about the table, and the results still look good. I'll maintain my score of 8.

---

### Official Review · Reviewer_rF97 · 2024-11-04

**Soundness:** 3
**Presentation:** 3
**Contribution:** 3
**Rating:** 6
**Confidence:** 4

**Summary:**

The paper introduces a benchmark to assess LLMs' ability to update their knowledge of API functions. It includes synthetic API function updates and program synthesis tasks using these updates. The paper evaluates various LLM update methods, such as fine-tuning open-source models on update documentation, prepending update information at inference time, and fine-tuning on examples. The dataset comprises 670 program synthesis examples involving updates to 54 functions from seven Python packages, all generated using GPT-4.

**Strengths:**

- **Relevant and timely research**: Updating APIs presents a significant challenge for LLMs. Therefore, assessing how effectively LLMs evolve with API changes addresses a valuable and necessary research area.
- **Novel techniques**:
    - The paper explores code-specific update methods like fine-tuning with examples, recognizing that LLMs must understand the semantics of an update rather than just its syntax.
   - The paper also establishes a new foundation for measuring LLM update capability in coding using UPass@k and specificity metrics.
- **Interesting insights for future research**: This paper also many interesting insights to help future research in this area. For example,
    - The paper shows that fine-tuning with examples outperforms fine-tuning on documentation.
    - The paper highlights the issue of specificity loss when fine-tuning with knowledge updates, encouraging future research to address this challenge.

**Weaknesses:**

**Evaluation limitations**: The model is penalized if it generates a correct program without using the updated function, even though it may still achieve the correct outcome. And from the examples given in the paper, it seems like this can occur very commonly. It would be helpful to explore a metric that captures these cases more accurately. **Suggestion**: Comparing pass@k with upass@k in Table 4 could offer some insight here.

**Possible compounding errors in LLM-generated datasets**: Since the dataset is fully generated by LLMs, LLMs can introduce errors affecting the quality of the dataset. It is very hard to establish that the update implementation is correct since the generated test cases themselves could be wrong. It is possible that LLM generated both wrong implementation and wrong test cases, but the wrong implementation might pass the wrong test cases. Similarly, LLM generated test cases for the program synthesis problems are used to the efficacy metric UPass@k and again there is no guarantee that these test cases are correct beyond just random manual inspection done by the authors.

**Questions:**

1.	Can you add pass@k metric to Table 4 in addition to UPass@k metric?
2.	How are LLMs updated on the docstring? For FT(U), how many times is the update instruction finetuned on? Does this hyper-parameter impact the performance?
3.	What are the random updates in the training examples in FT(PS) setting? Can you provide some examples?
4.	In Table 4, what is the baseline approach (is it the model without any fine-tuning or prompting for updates)? Additionally, what is the Upass@k for the baseline model? It’s surprising that this value is significantly greater than 0, given that models without update knowledge would likely fail to use the new API correctly.

---

> ### Author Response · Authors · 2024-11-26
> **Rebuttal**
>
> **[Weakness 1 / Question 1] The model may be unfairly penalized by UPass@k**
>
> Thanks for the suggestion! In our updated draft, we also report Pass@k in Table 4; this new metric measures the execution success with updated API in the environment. Under this metric, the model can pass tests without using the updated API. Unsurprisingly, Pass@k reports higher values for all models.
>
> In the following table, we address the reviewer’s concern by showing the model is NOT generating a correct program (without using updated function) __more than it does in the Base model__. We think the delta (**Pass@k - UPass@k**) reflects a certain number of problems where other solutions naturally exist. As we could see from the table, compared with Base, such gaps either remain on par, or shrink after (in-context or in-weight) updates. This implies that the updated models do not solve the problem without using updated function and thus we are not unfairly penalizing the model.
>
> |           |         | Pass@1 | UPass@1 | **Pass@1 - UPass@1** | Pass@5 | UPass@5 | **Pass@5 - UPass@5** |
> |-----------|---------|--------|---------|------------------|--------|---------|------------------|
> | GPT-4     | Base    | 54.1   | 2.7     | 51.4             | 74.5   | 5.7     | 68.8             |
> |           | Prepend | 63.9   | 34.1    | 29.8             | 83     | 57      | 26               |
> | Claude    | Base    | 51.8   | 2.9     | 48.9             | 61     | 3.6     | 57.4             |
> |           | Prepend | 68.4   | 58.7    | 9.7              | 77.2   | 71.9    | 5.3              |
> | CodeLlama | Base    | 28.4   | 4.4     | 24               | 39.4   | 7.6     | 31.8             |
> |           | Prepend | 32     | 6.7     | 25.3             | 44.6   | 10.6    | 34               |
> |           | FT (U)  | 28     | 4.3     | 23.7             | 40.9   | 7.3     | 33.6             |
> |           | FT (PS) | 28.6   | 22.9    | 5.7              | 45.7   | 37.6    | 8.1              |
> | DS-1      | Base    | 30.3   | 2.9     | 27.4             | 46.6   | 5.2     | 41.4             |
> |           | Prepend | 35.1   | 10.3    | 24.8             | 53.4   | 19.6    | 33.8             |
> |           | FT (U)  | 33.5   | 3.1     | 30.4             | 51.6   | 6.1     | 45.5             |
> |           | FT (PS) | 38.3   | 27.7    | 10.6             | 58.7   | 44      | 14.7             |
> | DS-1.5    | Base    | 46.8   | 3.2     | 43.6             | 64.3   | 6.4     | 57.9             |
> |           | Prepend | 50.9   | 11.8    | 39.1             | 70.7   | 22.1    | 48.6             |
> |           | FT (U)  | 47     | 3.6     | 43.4             | 65.4   | 7       | 58.4             |
> |           | FT (PS) | 38.7   | 29.4    | 9.3              | 61.3   | 47.2    | 14.1             |
>
>
> **[Weakness 2] Quality control for a LLM-generated dataset.**
>
> See general response.
>
> **[Question 2] How many gradient updates are performed when training on update docstring? And is it optimal?**
>
> In Table 4, we perform 10 gradient update steps on docstring (FT(U)) with a causal language modeling objective. We ran the FT(U) experiment in Table 4 with various gradient steps (see Table 10 in updated draft) and concluded the choice of 10 gradient updates is optimal. We include the table below for convenience.
>
> | DS-v1.5                    | UPass@1 | UPass@5 | SPass@1 | SPass@5 |
> |----------------------------|---------|---------|---------|---------|
> | FT (U) [10 gradient steps] | 3.6     | 7.0     | 56.4    | 77.3    |
> | FT (U) [5 gradient steps]  | 3.4     | 7.0     | 49.4    | 68.7    |
> | FT (U) [2 gradient steps]  | 3.4     | 7.0     | 49.5    | 71.3    |

---

> ### Author Response · Authors · 2024-11-26
> **Rebuttal (Continued)**
>
> **[Question 3] Example of random update in training examples in FT(PS) setting.**
>
> We give two examples as below:
> when training for update `numpy.argsort`, the target update has the following description:
>
> > Added a new parameter 'reverse' to the `numpy.argsort` function to support reverse sorting."
>
> Compare to a random example taken from elsewhere in the dataset:
> ```
> [random example to demonstrate task format]
> [INST] Scenario: Peter is a physics teacher and is developing an application for his students to calculate real-world problems. During a test, a student entered a negative number causing the application to break since square roots of negative real numbers do not exist. Peter now plans to update his application such that it doesn't break but instead returns a predefined result when a student enters a negative number.
> Problem: Peter wants you to write a function. This function should be designed in a way to handle both special and normal scenarios. In the usual situation, the function takes a real number and returns the square root of that number. However, in the special scenario, when students enter a negative number, instead of the program breaking due to an exception, it should return a predefined fallback number. The function must take three parameters: the first parameter is the number for which the square root has to be found, the second parameter is the fallback number and the third parameter will decide whether to use the fallback or not, if the first number is negative.
> Solution signature: def calculate_sqrt(number: float, fallback: float, use_fallback: bool)
> [TEST]
> …
> [/TEST]
> [/INST]
> ### Response:
> [PYTHON]
> import math
>
> def calculate_sqrt(number: float, fallback: float, use_fallback: bool):
>     # if use_fallback is set to True and number is less than 0, return the fallback value
>     if use_fallback and number < 0:
>         return fallback
>     # if use_fallback is not set and number is less than 0, let the math.sqrt function raise
>     # a ValueError as it would traditionally do
>     else:
>         # The new math.sqrt function signature takes in two parameters
>         # The first one is 'x' which is the number we need to find the square root of
>         # The second one is 'fallback' which is an optional parameter that defaults to None
>         # We use fallback parameter here to avoid raising an error
>         return math.sqrt(number, fallback)
>
> [/PYTHON]
> ```
>
> when training for update `torch.squeeze`, the target update has the following description:
>
> > Change the name of the argument 'dim' to 'dimension'.
>
> Compare to a random example taken from elsewhere in the dataset:
> ```
> [random example to demonstrate task format]
> [INST] Scenario: You're an engineer working in an AI based healthcare startup. Your team has been assigned with a task to analyze body scan images and extract diagonal arrays of pixel intensity values. These diagonal arrays will further be used to detect anomalies in human body scans.
> Problem: Given a 2D array representing the pixel intensity of a medical scan image and a shift 'k' from the leading diagonal, you need to write a Python function that extracts the given diagonal. These diagonals serve as features for your AI models, where different diagonals might signify different medical conditions.
> Solution signature: def extract_scan_diagonal(scan: numpy.array, shift: int)
> [TEST]
> …
> [/TEST]
> [/INST]
> ### Response:
> [PYTHON]
> import numpy
>
> def extract_scan_diagonal(scan: numpy.array, shift: int):
>     # Use the updated numpy API function
>     res = numpy.diag_extract_or_construct(scan, shift)
>
>     return res
>
> [/PYTHON]
> ```
>
> **[Question 4]  High performance of Base model**
>
> See general response.

---

> ### Author Response · Authors · 2024-12-03
> **Reminder about end of discussion period**
>
> Dear Reviewer,
>
> As we approach the response deadline, we wanted to kindly check if our revisions have adequately addressed your concerns. If you have any remaining questions or concerns, we would be happy to address them.
>
> Thank you for your time and consideration.
>
> Best regards,
>
> Authors

---

### Official Review · Reviewer_FSyK · 2024-11-04

**Soundness:** 2
**Presentation:** 2
**Contribution:** 1
**Rating:** 3
**Confidence:** 4

**Summary:**

This work proposes CodeUpdateArena, a benchmark for knowledge editing in the code domain. CodeUpdateArena evaluates the extent to which a pretrained LLM’s knowledge of code API functions can be updated. This benchmark is a set of API updates and program synthesis examples.

The authors evaluate different approaches for updating information, such as pretending the function update’s docstring in the prompt, fine-tuning on code update information, and fine-tuning on program synthesis examples. They find that fine-tuning on documentation of a new
update does not allow LLMs to incorporate changes for problem-solving, while prepending the same information or fine-tuning on examples does help the LLM.

**Strengths:**

- The paper is well-written, with an easy-to-follow running example in the description of the synthetic data generation process.
- The benchmark is open-sourced with 54 functions from seven diverse Python packages with 670 program synthesis examples.

**Weaknesses:**

- While useful, the impact of the paper is limited as the size and diversity of the data set is small (54 functions) in one programming language (python).
- The core contribution is that LoRA fine-tuning on natural language descriptions of code updates is worse than adding the updates in prepended prompts or LoRA fine-tuning on code examples.

**Questions:**

How might these results vary according to the size of the training dataset? E.g. if you employ full-parameter fine-tuning with significantly more samples, how might the efficacy of the FT methods change?

---

> ### Author Response · Authors · 2024-11-26
> **Rebuttal**
>
> **[Weakness 1] Limited size and diversity of the dataset**
>
> While we agree that other languages and libraries could be included, we believe that the number of functions is sufficient to support our claims. We sample these from different packages and with different types of update to cover the space of update types. For demonstrating superiority of one editing method over another, our benchmark is large enough to show this for any practically significant performance delta, assuming that the dataset reflects representative use cases.
>
> **[Weakness 2] Core contribution**
>
> We would like to emphasize that our core contribution is a dataset that provides a testbed for future researchers to investigate in teaching code LLM about package update and ensure it could use the update for problem solving beyond regurgitation.
>
> **[Question 1] How would experiment results change if the training dataset is significantly increased?**
>
> Past work like [1] shows what happens in a different regime where far more data is available.  Although our work supports multi-edit scenarios, we do not provide examples for conducting these types of experiments. We hope our work encourages future researchers to develop more sample-efficient algorithms to update code LLM methods. We believe this is important for the method to be scalable and deployed online (i.e., dealing with API updates from common libraries every day/week).
>
> [1] Physics of Language Models: Part 3.1, Knowledge Storage and Extraction

---

> ### Author Response · Authors · 2024-12-03
> **Reminder about end of discussion period**
>
> Dear Reviewer,
>
> As we approach the response deadline, we wanted to kindly check if our revisions have adequately addressed your concerns. If you have any remaining questions or concerns, we would be happy to address them.
>
> Thank you for your time and consideration.
>
> Best regards,
>
> Authors

---

### Official Review · Reviewer_PLK3 · 2024-11-04

**Soundness:** 3
**Presentation:** 4
**Contribution:** 2
**Rating:** 5
**Confidence:** 4

**Summary:**

This paper proposes a benchmark to assess a model's ability to adapt to changes in API function calling. They show that fine-tuning the model on updated program synthesis problems can improve the model's ability to embrace the new API function call.

**Strengths:**

This paper poses an interesting question of how to evaluate model's robustness to API changes and introduces a dataset to evaluate this. The further provide a testing environment to compare different approaches to augmenting model capabilities with new APIs, testing both in-context and fine-tuning approaches.

**Weaknesses:**

1) It's not clear that the types of updates proposed by GPT-4 are representative of the types of updates found in the wild. With synthetic data, GPT-4 may be idiosyncratic in the updates proposed, step-4 of deduplication removing 53% of the problems seem to indicate this is a significant downside of using GPT-4 for problem generation. A comparison to historic API changes would help justify this.

2) Related work would be better in section 2 to provide better context on what other work has been done in this topic.

3) The naming of "Arena" seems to imply side-by-side comparison and human input the way ChatbotArena does. This benchmark does not do this. The creation of test cases and sample tasks that involve the new updated API function is more akin to R2E https://r2e.dev/pdfs/paper.pdf

4) It's not clear that updates to APIs happen at a high frequency compared to updates to post-training in proprietary models. In those cases, it may be simper or beneficial to amend the data in the post-training corpus. It would be worth studying if models can be taught to distinguish the difference between package/api versions.

5) The results are limited to statistical and data processing tools popular in python. It would be interesting to show on other APIs like web applications and visualization packages like plotly and matplotlib.

**Questions:**

1. Do these results replicate when not using LoRA but using full fine-tuning? It's possible that full fine-tuning with data augmentation will allow the model to pay better attention to the in context docstrings and API description. Perhaps something like RAFT (https://arxiv.org/abs/2403.10131).

2. Does GPT-4 fine-tuning lead to improved performance? Otherwise, this result might be only limited to smaller models with weak reasoning and coding abilities.

3. Does the fine-tuning lead to improved capabilities for other coding tasks like test-generation for code, self-repair, and code execution?

---

> ### Author Response · Authors · 2024-11-26
> **Rebuttal**
>
> **[Weakness 1] Effort to make synthetic API reflect realism**
>
>
> See general response.
>
> **[Weakness 2] Related work positioning.**
>
> We have updated the draft and moved the related work section; thanks for the suggestion.
>
> **[Weakness 3] Naming the benchmark “Arena”**
>
> While Chatbot Arena is the most well-known benchmark at the moment, multiple prior work (LongRangeArena (https://arxiv.org/abs/2011.04006), WebArena (https://arxiv.org/abs/2307.13854), Arena (https://ojs.aaai.org/index.php/AAAI/article/view/6216) used “Arena” to refer to a place to test the model (which reflects our use case) without invoking side-by-side comparison and human input.
>
> **[Weakness 5] Extend synthetic API generation to APIs like web applications and visualization packages**
>
> We mention this in our limitations paragraph. We foresee that how to evaluate the correctness of such scenarios automatically is a chief barrier to doing this, with no good solution easily available.
>
> **[Question 1] LoRA Finetuning (our work) vs Fully-Fine Tuning (FFT) with data augmentation**
>
> In our preliminary experiment on a different dataset, we found LoRA outperforms full fine-tuning (FFT).  Our use of LoRA follows that of other recent editing work like [1]. We chose to use LoRA for our experiments as it is substantially more computationally efficient than full fine-tuning (10x). Nevertheless, we believe that sufficient data augmentation compared with full fine-tuning, like RAFT, could lead to improved performance.
>
> [1] Gangadhar and Stratos. “Model Editing by Standard Fine-Tuning”. arXiv 2024.
>
> **[Question 2] Whether updated model becomes stronger in other capabilities**
>
> Since our core contribution is the dataset that opens a new domain for code generation, we think the answer to this intriguing question is left for future work. To our best guess, finetuning would not lead to improved capabilities, as the learning itself doesn’t involve teaching the model more things beyond the target update. Also, the vanilla finetuning algorithm doesn’t “propagate” new knowledge well — finetuning the model on docstrings doesn’t teach the model to solve related program synthesis problems.
>
> **[Question 3] The base model we chose to conduct the experiment is not strong enough.**
>
> We respectfully disagree with this point. We found DS-Coder-Instruct-7B is the strongest open model (at comparable size) so far; according to their paper (https://arxiv.org/pdf/2401.14196), it surpasses GPT-3.5 Turbo and are only less than GPT-4 by 6% on HumanEval (Python). Therefore, what we observed in DS-Coder-Instruct-7B may be reflective of what might happen to a strong code model like GPT-4.

---

> ### Author Response · Authors · 2024-12-03
> **Reminder about end of discussion period**
>
> Dear Reviewer,
>
> As we approach the response deadline, we wanted to kindly check if our revisions have adequately addressed your concerns. If you have any remaining questions or concerns, we would be happy to address them.
>
> Thank you for your time and consideration.
>
> Best regards,
>
> Authors

---

### Official Review · Reviewer_FeEE · 2024-11-06

**Soundness:** 2
**Presentation:** 3
**Contribution:** 2
**Rating:** 5
**Confidence:** 5

**Summary:**

This paper proposes CodeUpdateArena, a dataset of synthetic updates to library APIs in Python. Each instance in the dataset consists of the update (e.g., add/modify a parameter), a code synthesis problem using the changed API, and a set of tests to validate the synthesis results. The authors use GPT-4 to generate most of the dataset, followed by manual inspection to eliminate low-quality or incorrect instances.

The authors explore three evaluation strategies with closed-source and open-source LLMs (GPT-4, Claude, DeepSeekCoder, and CodeLLama): prepending prompts with api update information, fine-tuning with docstrings, and fine-tuning with synthesis examples.

The results show that the prepend and fine-tuning options improve the efficiency of the LLMs but also hurt their specificity, i.e., their performance on unrelated tasks.

**Strengths:**

- The paper targets an interesting problem about the integration of API updates in LLM code generation. I believe this is an important task given that LLMs do not have knowledge about the version of the libraries that they use to generate code. Hence, without the updated information, the generated code might be incorrect or not compilable.

- The proposed dataset, CodeUpdateArena, is potentially useful to evaluate if LLMs can generate updated information.

- The paper covers a reasonable set of LLMs in their evaluation, including both closed-source and open-source LLMs.

**Weaknesses:**

- The fact that dataset is generated using GPT-4, including both the updates, synthesis problems, and the tests to evaluate the generated synthesis, is questionable. This approach limits the quality and realistic nature of the dataset. There is historical data and commits on API updates for many libraries on Github -- which would be more realistic and perhaps still challenging for the LLM to work with.

- Unlike historical facts, multiple versions of a library API can exist at the same time and are concurrently used by different users due to compatibility issues. E.g., the latest version of numpy may require python 3.12 or greater; such users may want the older API. Hence, getting the latest version of the API is not always important, rather the LLM may have to understand the context or dependencies better for correct prediction. Hence, fine-tuning the LLM with fixed updates does not make sense.

- Further, numerous libraries are updated every day/week, hence fine-tuning LLMs every time for these updates is not scalable.

- The evaluation approach and metrics do not provide a way to gain high confidence if the LLM learned the API update. There can be many other synthesis scenarios that requires using the API in question. How can we confidently say if the LLM learned the update if the evaluation is only done on a handful of synthesis examples? Hence a more robust evaluation is needed for this aspect.

- Given the fact that the larger LLMs like GPT-4 and Claude already obtain ~60-70% efficacy, this benchmark seems to be challenging only for smaller and slightly outdated LLMs. With the latest LLMs, the benchmark might soon saturate. Hence, it is unclear if the benchmark will be useful to evaluate the upcoming generations of LLMs.

**Questions:**

1. Why do the authors generate synthetic data instead of using historical API updates?
2. How can we rely on tests generated by the LLM?
3. Given the very few examples, the SPass metric is a weak measure of LLM's updating ability? How can we gain more confidence in this metric?

---

> ### Author Response · Authors · 2024-11-26
> **Rebuttal**
>
> **[Weakness 1 / Question 1] Justification of synthetic data instead of using historical API updates.**
>
> See general response.
>
> **[Weakness 2] Multiple versions of API can be useful for users**
>
> Thank you for bringing this point. However, we don’t see this as a weakness impacting our work. The current dataset tests the ability to overwrite an old API with a newer version. One could use this capability to inject knowledge associated with a particular library version. A knowledge injection method that works on our dataset can generalize to that use case as well.
>
> **[Weakness 3] Scalability of updating LLMs for each update**
>
> Since our benchmark contains multiple updates for multiple libraries, future researchers can use our dataset to investigate techniques that update multiple APIs at the same time or in sequential order (at least, on a subset of examples in our dataset that do not update the same function).
>
>
> **[Weakness 4 / Question 3] Robustness of evaluation metric**
>
> The robustness of our observation from result tables comes in expectation over the many synthesis examples of many updates — we test on 82 HumanEval examples (a quantity larger than prior work [1,2]), after we finish updating the model on each API update. We note that this is a common practice in other benchmarks; for example, RAG benchmarks don’t involve exhaustively answering questions about the entire content of a document. Additionally, we confirm such a difference with a bootstrap significance test with p < 0.05.
>
> [1] Locating and Editing Factual Associations in GPT, Meng et. al, NeurIPS, 2022
>
> [2] Propagating Knowledge Updates to LMs Through Distillation, Padmanabhan et. al, NeurIPS, 2023
>
>
> **[Weakness 5] Headroom of the benchmark**
>
> The goal of our benchmark is to test updating methods, not specific models. In this work, our LLMs like GPT-4 and Claude are prompted through RAG, which doesn’t do in-weight parameter updates. These models aren’t addressing the task at hand, so the high performance numbers have no bearing on knowledge updating.
>
> **[Question 2] Quality control for an LLM-generated dataset**
>
> See general response.

---

> ### Author Response · Authors · 2024-12-03
> **Reminder about end of discussion period**
>
> Dear Reviewer,
>
> As we approach the response deadline, we wanted to kindly check if our revisions have adequately addressed your concerns. If you have any remaining questions or concerns, we would be happy to address them.
>
> Thank you for your time and consideration.
>
> Best regards,
>
> Authors

---

> > ### Comment · Reviewer_FeEE · 2024-12-03
> > **Response to rebuttal**
> >
> > Thanks for the responses. Some of my major concerns regarding validity and realism still remain after reading the response. I would have liked to see some manual validation of the dataset, given its small size -- this would improve the reliability of the dataset and encourage LLM developers to use it. However, the dataset might still be somewhat useful for evaluating LLMs. Hence, I am slightly increasing the score.
> >
> > [W1]: I am still not convinced by this response. To avoid data leakage, one can always choose updates after the latest training date for the LLMs. However, this is not a major issue. The main problem is the realism of the dataset. While the taxonomy is useful, it is unclear whether it is comprehensive or diverse enough. Adapting updates from historical data might mitigate this problem and also generate more challenging scenarios.
> >
> > [Q2]: Its good that authors implement quality control steps. However, the main concern about correctness of the updates and tests still remains. Incorrect solutions can still remain after the simple filtering steps.
> >
> > [W3]: "future researchers can use our dataset to investigate techniques that update multiple APIs at the same time" -- this needs to be evaluated, how can one know whether this works at all? So scalability problems will still remain.
> >
> > [W2]: I agree that this is not a problem for the dataset, but it is a problem for the rest of the evaluation involving fine-tuning and other methods.

---

### Author Response · Authors · 2024-11-26
**General Response**

# General Response Part 1: High performance of base model

Reviewer **rF97** and **jDUt** note that the Base model achieves high UPass@k performance without knowing updated API information. We thank the reviewer for raising this good point.

Our unit tests relied on calling APIs themselves to produce the correct answer. The unit tests were themselves correct, and correctly assessed model performance under new API updates, but this had the effect of causing failures in the unit tests when running with the old API. To obtain a high UPass@k, the predicted solution needs to fail unit tests with the old API in the environment (as an indicator of using updated API), so this instant failure leads to a high false positive of UPass@k.

**To address this, we converted the answers of the unit tests into literal values and reran all impacted numbers.** The changes are as follows:
1. UPass numbers are substantially lower than before, particularly for the baselines. However, the deltas between approaches (e.g., Baseline and Prepend) are roughly the same as before, and the method that works the best (FT (PS)) is unchanged.
2. Pass@k numbers (added in Table 4) are slightly different due to changes in the unit tests (e.g., Pass@1 for GPT-4 baseline was 54.0% and is now 54.1%; Pass@1 with Claude was 69.5% and is now 68.4%).

We have included Appendix B.4 describing this change. Table 7 includes the results from the previous version of the PDF before the change was made.


# General Response Part 2: Dataset Realism
Reviewer **FeEE** and **PLK3** comment on our choice of using synthetic data for the study from two aspects: 1) synthetic data may not reflect real world API updates; and 2) using historical API updates is a plausible alternative.

We chose synthetic data construction primarily to prevent data contamination, since code LLMs are pretrained on large code corpora and recent pre-training corpora would contain historical API usage info. We see this as a fundamental weakness for a dataset studying knowledge editing methods. Furthermore, real API updates often entangle a large number of changes; for a benchmark dataset on knowledge editing, real updates do not provide the level of modularity needed for focused evaluation.

To make sure our generated updates are as realistic as possible, we built a taxonomy (Line 813-830), and human spot checks for updates during generation (Line 263-268). Our data was intentionally designed to mirror the types of updates seen in the wild. We agree that comparing actual historical data would be an interesting study, though we are not aware of off-the-shelf datasets of past API updates.

# General Response Part 3: Quality Control of an LLM-Generated dataset

Reviewer **FeEE** and **rF97** expressed concern about the quality of an LLM-generated dataset. We address the concern here.

We ensure the correctness through systematic design of our generation pipeline. We generate different parts of material (e.g. update description, update implementation) in a cascading fashion. Taking unit tests as an example,  we have generated each unit test and different sections of each unit test — input, answer, assertion — in separate calls to GPT-4 (middle part of Figure 2). This allows self-consistency [1] to come into play. If the model agrees with itself (measured by executing solution) beyond a certain threshold (we set it to be 60%-70% Line 258, Line 300), it’s likely the case that the entire bundle — problem specification, reference solution, unit tests, etc — is correct.

We further ensured this by conducting multiple quality control practices during development: spot checking generated update (Line 263-Line 268), deduplication of program synthesis examples (Line 304-309), inspection of failed GPT-4 generation (Line 341-348, Line 1757-1765, Table 9), and verifying test coverage (Line 1816-1820).

We further note that there are examples in the community where imperfect, automatically generated benchmarks are still useful. For example, post-training methods are tested on LLM-generated datasets — UltraFeedback, AlpacaFarm [4,5]; CodeBenchGen [2], an LLM-generated benchmark for code generation; MuSR [3], an LLM-generated benchmark for testing reasoning, etc.

[1] Self-Consistency Improves Chain of Thought Reasoning in Language Models

[2] CodeBenchGen: Creating Scalable Execution-based Code Generation Benchmarks

[3] MuSR: Testing the Limits of Chain-of-thought with Multistep Soft Reasoning (ICLR 2024)

[4] UltraFeedback: Boosting Language Models with Scaled AI Feedback

[5] AlpacaFarm: A Simulation Framework for Methods that Learn from Human Feedback

---

### Meta-Review · Area_Chair_haTa · 2024-12-19

**Metareview:**

The paper investigates a practical problem: updating a code language model to be aware of a change to API functions. They find prompting is surprisingly more effective than fine tuning on a new synthetic dataset that they introduced generated via prompting GPT4. It is best seen as a benchmarking and evaluation paper.

The primary strength is that the problem is important. The primary weakness is that the dataset itself is not rich enough to serve as a foundation for organizing the community around this problem: It is already reasonably well handled by frontier LLMs (unsurprising, given that it was generated by them), and so likely to saturate soon, and although there is some spot checking of solutions it is not as compelling as a human created benchmark (cf sygus), and lack robust automatic validation of code correctness for anyone who wanted to attempt the benchmark.

Given that this is primarily a benchmark paper it is hard to argue for acceptance when there are important problems with the benchmark itself, and little methodological novelty. Therefore this paper should be rejected.

**Additional Comments On Reviewer Discussion:**

The papers champion says "a novel approach to solving the problem isn't presented" and that this is more of a benchmark paper. But the papers detractor FeEE raised important issues with the paper from a benchmark perspective, which were not rebutted by the authors apart from saying that they spot check the solutions, but this is more about rebutting an objection to the baseline numbers quoted, when the core issue was with the benchmark itself.

---

### Decision · Program_Chairs · 2025-01-22

Reject